# Diet and Nutrition Status of Mongolian Adults

**DOI:** 10.3390/nu12051514

**Published:** 2020-05-22

**Authors:** Sabri Bromage, Tselmen Daria, Rebecca L. Lander, Soninkhishig Tsolmon, Lisa A. Houghton, Enkhjargal Tserennadmid, Nyamjargal Gombo, Rosalind S. Gibson, Davaasambuu Ganmaa

**Affiliations:** 1Department of Nutrition, Harvard T.H. Chan School of Public Health, Boston, MA 02115, USA; 2Department of Neurology, Ulm University, Ulm 89081, Germany, and Central Scientific Laboratory, Institute of Medical Sciences, Ulaanbaatar 16081, Mongolia; tsemena312@gmail.com; 3Department of Pediatrics, Section of Nutrition, University of Colorado School of Medicine, Aurora, CO 80045, USA; rebecca.lander3@gmail.com; 4School of Public Health, Mongolian National University of Medical Sciences, Ulaanbaatar 14210, Mongolia; soninkhishig@must.edu.mn; 5Department of Human Nutrition, University of Otago, Dunedin 9054, New Zealand; lisa.houghton@otago.ac.nz (L.A.H.); rosalind.gibson@otago.ac.nz (R.S.G.); 6Nutrition Laboratory, National Center for Public Health, Ulaanbaatar 13381, Mongolia; enke98@yahoo.com; 7Food and Agriculture Organization of the United Nations Country Office, Ulaanbaatar 14201, Mongolia; nyamjargal.gombo@fao.org; 8Department of Nutrition, Harvard T.H. Chan School of Public Health and Channing Division of Network Medicine, Department of Medicine, Brigham and Women’s Hospital and Harvard Medical School, Boston, MA 02115, USA

**Keywords:** nutrition assessment, diet survey, dietary pattern analysis, nutrient inadequacy, overweight and obesity, nutritional epidemiology, nutrition transition, pastoral nomadism, Mongolia, central Asia

## Abstract

(1) Background: Aspects of the Mongolian food supply, including high availability of animal-source foods and few plant foods, are plausibly associated with disease in the population. Data on Mongolian diets are lacking, and these risks are poorly quantified. The purpose of this study was to provide a multifaceted nutritional analysis of the modern Mongolian diet. (2) Methods: The study population consisted of 167 male and 167 female healthy non-pregnant urban and nomadic adults (22–55 years) randomly selected from lists of residents in 8 regions. From 2011–2016, 3-day weighed diet records and serum were collected twice from each participant in summer and winter; anthropometry was collected once from each participant. Serum was analyzed for biomarkers, and nutrient intake computed using purpose-built food composition data and adjusted for within-person variation. Exploratory dietary patterns were derived and analyzed for associations with diet and nutrition measurements. (3) Results: We collected 1838 of an expected 1986 diet records (92.5%), 610/658 serum samples (92.7%), and 315/334 height and weight measurements (94.3%). Sixty-one percent of men and 51% of women were overweight or obese. Consumption of red meat, refined grains, and whole-fat dairy was high, while that of fruits, non-tuberous vegetables, eggs, nuts and seeds, fish and poultry, and whole grains was low. Dairy and red meat were more consumed in summer and winter, respectively. Dietary inadequacy of 10 of 21 assessed nutrients, including fiber, folate, and vitamin D were >50% prevalent, while protein, zinc, and vitamin B12 inadequacy were low. Biochemical evidence of iron and vitamin A deficiency was also low. Three dietary patterns (Urban, Transitional, Nomadic) explained 41% of variation in food consumption. The Urban pattern was positively associated with BMI in multivariate analysis. (4) Conclusions: Results indicate a high prevalence of key dietary inadequacies and overweight among Mongolian adults. Prior studies by our group have suggested that expanded supplementation and food fortification would be effective in addressing micronutrient inadequacies; these strategies should be coupled with measures to mitigate the growing burden of chronic disease.

## 1. Introduction

Due to climatic factors and a traditionally nomadic lifestyle, Mongolia contains a smaller fraction of arable land devoted to permanent crops than any other country (0.003%) [1], and the national food supply is marked by a pattern of extremes. Normalized against the total per capita supply of caloric energy in each country, data from the Food and Agriculture Organization of the United Nations (FAO) from the most recent year available indicate that the daily per capita food supply of Mongolia ranked 9th and 12th out of 175 countries in terms of whole milk and meat supplied, respectively, and ranked in the bottom 10% of countries in the supply of fruit, fish and seafood, pulses, and oil crops [2]. Particularly in rural areas, food consumption remains highly seasonal, especially with respect to dairy products and meat which are mostly consumed in summer and winter, respectively [3].

Extremes in Mongolia’s national food supply are plausibly related to the national burden of disease, particularly chronic disease. Recent analysis of the Global Dietary Database shows that Mongolia scored lower than any other country in both 1990 and 2017 in the Alternative Healthy Eating Index [4]. Using FAO food balance data to estimate food consumption in Mongolia [5], the Global Burden of Disease study found that out of 195 countries and territories in 2017, Mongolia ranked 1st globally in the fraction of cardiovascular (CVD) mortality attributable to dietary imbalances in both men (62.8%) and women (60.5%) and 12th in the rate of age-standardized all-cause mortality attributable to diet (309 deaths per 100,000) [6]. Mongolia also ranked 22nd and 12th in rates of all-cause and CVD mortality attributable to metabolic risk factors. While childhood stunting and vitamin A deficiency have decreased over the past two decades, this period has also seen a sharp rise in the prevalence of adolescent and adult overweight and type 2 diabetes [7,8]. Currently, almost one third of pregnant women are anemic, less than two thirds of infants are reportedly exclusively breastfed up to six months of age, and vitamin D deficiency is endemic [7,9].

Despite these concerning statistics, national assessments of dietary intakes of individuals are lacking for the Mongolian population. From 1999–2009, dietary assessments of individuals and analytic epidemiologic research on Mongolian and Inner Mongolian diets were pioneered by Mongolian and Japanese scientists [10,11,12,13,14,15], but methods used in these studies were not applied on a nationwide scale. Current nationally-representative food or nutrient consumption data at the individual level are not available. Without such data, it is impossible to precisely quantify the extent of dietary imbalances throughout the population, and difficult to design evidence-based and targeted strategies to address such imbalances.

The objective of this study was to provide a multifaceted nutritional analysis of the modern Mongolian diet, using paired summer and winter weighed diet records collected from urban and rural Mongolian men and women living in 8 regions of the country. As part of a prior analysis based on these diet records [16], we reported the extent of population dietary inadequacy of 10 micronutrients; the present analysis extends this investigation to 21 nutrients, includes analysis of daily intake of food groups and their contributions to nutrient intakes, and includes exploratory analysis of population diet patterns. To support interpretation of the dietary data, we also assessed anthropometry and selected micronutrient biomarkers.

## 2. Materials and Methods

### 2.1. Study Population and Data Collection

The study population consisted of 167 male and 167 female healthy non-pregnant adults aged 22–55 and living in separate households. Eligible participants were randomly sampled from lists of urban and peri-urban residents of the capital Ulaanbaatar and the provincial centers of 7 aimags (provinces) spanning the country (Bulgan, Dornod, Khuvsgul, Khovd, Omnogobi, Sukhbaatar, and Tuv), and nomads residing in soums (rural districts) within a short drive from each provincial center (Archon, Bayantumen, Chagall, Buyant, Bayandalai, Khalzan, and Altanbulag, respectively). Participants were located and followed-up with the aid of local health officers (Figure 1).

From 2011 to 2016, 3 consecutive days of weighed diet records (including 1 weekend day, and including assessment of portions consumed as well as ingredient composition of mixed dishes), were collected from each participant by dedicated research assistants in both summer (June–August) and winter (December–February). In addition, serum samples were collected from each participant during both seasons, and height and weight measured in the winter only for the majority of participants. As fasting is not required for accurate measurement of the biomarkers of interest, serum was collected at random times during the day in order to avoid systematic error in dietary measurements that could occur from interrupting all subjects at the same time (e.g., before breakfast). Details of the dietary assessment, blood collection, and anthropometry have been described earlier [9,17].

Procedures followed were in accordance with the ethical standards of the Mongolian Ministry of Health Ethical Review Board and the Harvard T.H. Chan School of Public Health Institutional Review Board (Protocol 21002). Eligible participants provided written informed consent prior to enrolment and were free to withdraw from the study at any time.

### 2.2. Analysis of Anthropometric and Micronutrient Status

Body mass index was calculated using participants’ height and weight, and categorized according to WHO cutoffs [18] based on evidence suggesting that these are more appropriate than WPRO Asian-specific cutoffs for use in Mongolia [19]. Serum was analyzed for ferritin, soluble transferrin receptor (sTfR), retinol binding protein (RBP), standard C-reactive protein (CRP, a marker of acute-phase systemic inflammation), and alpha-1-acid glycoprotein (AGP, a marker of chronic inflammation) using a validated sandwich ELISA [20]. Iron and vitamin A deficiency were defined according to serum ferritin and RBP concentration cutoffs, respectively, differentially adjusted for inflammatory defined according to different combinations of elevated CRP (5 mg/L) and AGP (1 g/L): normal-, incubation/late convalescence-, and early convalescence-specific cutoffs used for ferritin and RBP were 15/19/27 µg/L and 0.7/0.6/0.5 µmol/L, respectively [21]. Iron overload was defined as serum ferritin >300 ng/mL in men and >200 ng/mL in women without the presence of inflammation [22].

Means and prevalence of categories of BMI and biomarkers were calculated within subgroups (defined by urbanicity and sex) and season, as well as within regions in both seasons. Significant differences (*p* < 0.05) were identified for the following planned comparisons: summer vs. winter within urbanicity-sex, males vs. females within urbanicity (BMI) or urbanicity-season (serum biomarkers), urban vs. rural within sex (BMI) or sex-season, and each province vs. the mean of all other provinces. Least squares analysis was conducted to identify marginal effects of urbanicity, sex, and age on mean biomarkers, adjusting for covariates. Statistical analyses were performed in R version 3.4.3 (R Foundation for Statistical Computing, Vienna, Austria).

### 2.3. Analysis of Diet Records

Empirical recipes and dish yields [23] were generated for complex dishes by averaging information on the weight of raw ingredients versus cooked dishes when available, or by applying ingredient yield factors [24,25]. Nutrient composition of ingredients, single-ingredient food items, and complex dishes were compiled using a combination of unpublished locally-analyzed food composition data from the Mongolian University of Science and Technology and Mongolian Public Health Institute, food composition data from the United States and Germany [26,27], and entries from a combination of international food composition tables previously compiled as part of a food composition table for Mongolian children [28]. Where applicable, nutrient retention factors [24,29,30] were applied to calculate nutrient concentrations in cooked foods, and adjustments made, where necessary, to borrowed nutrient values for differences between the moisture and fat content of the Mongolian and borrowed foods [23].

Usual intake distributions of nutrients and nutritionally-relevant food groups were estimated for each of the 8 subgroup-seasons (e.g., Urban Males in Summer) and each of the 8 regions, using the Statistical Program to Assess Dietary Exposure (SPADE) in R [31], which corrects observed measurements for within-person variation using variance components estimated from the study population. For each food group or nutrient, the estimation method used in each subgroup-season was dependent on the observed frequency of consumption as follows (Appendix A):(1)Distribution of habitually-consumed components (those observed to be consumed on at least 95% of diet record days, including some food groups and all nutrients except alcohol) was estimated using a model that estimates intake amounts only.(2)Distribution of episodically-consumed components (those consumed on less than 95% of diet record days, including alcohol and most food groups) was estimated using a model that estimates both intake frequencies and intake amounts.(3)In cases where consumption of a food group was too infrequent for the episodic model to produce an intake distribution, mean intake was calculated by computing each person’s average daily intake in each season and then averaging within-person seasonal means across persons in each subgroup-season.

Estimated distributions were used to derive mean intake of foods and nutrients, prevalence of nutrient inadequacy, and prevalence of intakes above the upper limit for all participants in each stratum as well as by tertiles of age (<33, 33–44, 45+ years). Statistics were weighted according to the distribution of weekdays and weekend days. A full-probability approach was used to estimate inadequacy for all nutrients whose requirement was assumed to be normally distributed with a coefficient of variance (CV) of 10% except for protein (CV = 12.5%), vitamin A (20%), and niacin and copper (15%) [32]; for iron among women of reproductive age, a lognormal requirement distribution was assumed. Given the population’s high observed meat consumption and low phytate:zinc molar ratio, EARs from the U.S. National Academy of Medicine were considered appropriate [33].

The following quantities were also calculated in each stratum using the sequential averaging procedure described in (3) above:(1)For each food group, mean daily intake density (g/2500 kcal/day, where 2500 kcal was approximately equal to the grand mean daily energy intake in the study population) was calculated to aid comparison of food intake across strata with varying levels of energy intake.(2)A separate set of “dish-based” food groups (distinguished by local culinary practices instead of inherent nutritional significance) were analyzed for their mean proportional contributions to total daily intake of each nutrient.(3)The mean fractional contribution of protein (4 kcal/g), carbohydrate (4 kcal/g), fat (9 kcal/g), and alcohol (7 kcal/g) to total energy intake; mean fraction of vitamin A contributed by retinol; and phytate:calcium, phytate:iron, and phytate:zinc molar ratios were calculated.

Exploratory diet patterns were analyzed using the “prcomp” package in R. Patterns were generated based on summer and winter within-person daily means of 12 food groups selected as major contributors of energy and nutrient intake in the study population. Patterns were retained according to quantitative criteria (variance explained by each rotated factor, and all factors as a whole, based on their associated eigenvalues and inspection of a scree plot) and qualitative criteria (interpretability of each factor with respect to its combination of high and low factor loadings, and the distribution of adherence to each pattern across population subgroups). Factor loadings were used to calculate season-specific patterns scores for each participant, and scaled from 0–100. Estimated marginal means of diet pattern scores were compared by age group, and selected nutrient intakes and biomarker measurements were compared across quintiles of diet pattern scores.

## 3. Results

Based on a possible maximum of 6 diet records (3 per season), 2 serum samples (1 per season), and 1 height and weight measurement collected from each participant, participants (167 men and 167 women) completed 1838 of 1986 (92.5%) diet records (mean number of records per person: 5.50), provided 610 of 658 (92.7%) serum samples (mean number of samples per person: 1.83), and 315 of 334 (94.3%) height and weight measurements (Appendix A). Mean age of participants was 39.2 years (Appendix A). Additional population characteristics have been reported previously [9]. There were no significant differences in age, sex, urbanicity, province, or season between participants who provided a serum sample in both summer and winter and 3.9% who provided only one sample.

Urban men and women consumed more fruits and non-tuberous vegetables than their rural counterparts in both seasons, with urban women consuming markedly more fruit than any other population subgroup (62 and 69 g/day in summer and winter, respectively) (Table 1). However, consumption of fruits, non-tuberous vegetables, eggs, nuts and seeds, fish and poultry, and whole grains was generally low across population subgroups and seasons, and consumption of deep orange tubers was not observed. Total consumption of milk and dairy products ranged from 116 g/day among urban males in winter to 657 g/day among rural males in summer, with rural men and women consuming more than their urban counterparts in both seasons, and summer consumption exceeding that in winter among all subgroups. Consumption of reduced-fat milk or dairy products was not observed. Consumption of juice and sugar-sweetened beverages (SSBs) was highest in urban areas, particularly among urban males (178 and 138 g/day in summer and winter, respectively), while consumption of sweets was highest among urban females (28 and 34 g/day in summer and winter, respectively), and approximately double that of their rural counterparts. Consumption of meat was extremely high, particularly among men and in winter (with urban and rural men consuming 425 and 450 g/day in winter, respectively), as was that of refined grains (ranging from 322 g/day among rural females in summer and 533 g/day among rural males in winter). Age-trends in food consumption are presented in Appendix A.

Across subgroups and seasons, protein, carbohydrates, fat, and alcohol contributed an average of 21%, 40%, 38%, and 1% of caloric energy, respectively, and mean phytate intake did not exceed 500 mg (Table 2). The prevalence of dietary protein, copper, phosphorous, zinc, riboflavin, niacin, and vitamin B12 inadequacy was very low, only exceeding 10% for riboflavin and niacin among urban females in winter. A moderate prevalence of iron inadequacy was observed among urban women (19 and 21% in summer and winter, respectively) and rural women (16 and 14%). Inadequacies of calcium, magnesium, thiamin, and vitamins A and B6 were common, and almost the entire study population was inadequate in folate and vitamins C, D, and E in both seasons. Median intakes of dietary fiber and potassium fell short of sex-specific adequate intake levels, and median intakes of manganese and pantothenic acid generally met or exceeded adequate levels (median intake (IQR) of fiber, potassium, manganese, and pantothenic acid averaged across the 8 subgroup-seasons: 9.3 (4.3), 2732 (1117), 3.176 (1.568), 6.354 (2.596), respectively). Urban-rural differences in nutrient inadequacy were most salient for calcium (more commonly inadequate in urban areas), vitamin A (more common in urban areas in summer, and rural areas in winter), and vitamin B6 (more common in rural areas in summer). The prevalence of intakes above the upper limit did not exceed 10% except for calcium among rural males in summer (12%) (Appendix A), due to their high summer intake of milk and dairy products (mean: 657 g/day) which contributed 62% of their total calcium intake in that season. Age-trends in nutrient intake and adequacy are presented in Appendix A.

Dishes contributing the majority of dietary energy in urban and rural areas included bread with or without toppings (contributing an average of 10% across urban and rural men and women in summer and winter, and 8% across rural subgroups), milk, dairy products, and airag (fermented mares’ milk) (10 and 18% in urban and rural subgroups respectively), biscuits, cookies, and doughnuts (12 and 18%), stir fries (21 and 15%), soups (14 and 13%), dumplings not included in soup (14 and 13%), and miscellaneous meat dishes (9 and 10%) (Table 3). See Appendix A for each dish’s contribution to other nutrients.

Of the study population, 61% of the men and 51% of the women were overweight, with 16% and 8%, respectively, being obese; 3% were underweight (Table 4). Mean BMI was significantly higher among men than women in both urban areas (men: 26.9 kg/m^2^, women: 25.5) and rural areas (men: 26.0, women: 24.6), and did not differ across urban and rural areas within sex. In adjusted analyses, urban residence was independently associated with higher BMI (adjusted mean: 26.4 vs. 24.9 in urban and rural areas, respectively) (Table 5). Biochemical iron deficiency, iron overload, and vitamin A deficiency were observed only in *n* = 12, 4, and 2 samples, respectively, and elevated CRP and AGP concentrations were observed in 11 and 7% of samples, respectively (Table 4). Urban residence was independently associated with higher serum RBP and lower AGP concentrations (Table 5).

Factor loadings, and the mean observed intake of each factor component across quintiles of factor scores in both seasons combined are presented in Table 6 and Figure 2. Three patterns were retained: an “Urban” pattern (attributing 21% of variance in intake of factor components) marked by high consumption of vegetables, juice and sugar-sweetened beverages, liquid oils, red meat, refined grains, and white roots and tubers; a “Transitional” pattern (11% of variance) marked by high consumption of dairy products, sweets, and fruit, and low alcohol and red meat; and a “Nomadic” pattern (10% of variance) marked by high consumption of dairy products, milk, red meat, and refined grains, and low juice and SSBs, processed meat, and fruit. In adjusted analyses, adherence to the Urban pattern was significantly higher among urban and Ulaanbaatar residents; the Transitional pattern was most associated with female sex, and the Nomadic pattern with rural residence (Table 7). Urban pattern scores were independently associated with younger age in all subgroups except rural females, the Transitional pattern was not associated with age in any subgroup, and Nomadic pattern scores increased with age among urban and rural males (Table 8, Figure 3). Adjusting for total energy intake and other covariates, increased adherence to the Urban pattern was significantly associated with increased intakes of protein, fiber, iron, and zinc, and decreased calcium intake; the Transitional pattern with increased protein intake; and both the Transitional and Nomadic patterns with increased intakes of iron and zinc, and decreased fiber intake (Table 9).

Trends in estimated marginal means of energy intake and body mass index by age and pattern scores across subgroups are presented graphically in Figure 4. Despite no significant differences in linear trend of adjusted energy intake with age between any population subgroups, a significant upward trend in BMI with age was observed among both men and women in urban areas but not rural areas (Figure 4, Table 5). Increased adherence to all pattern scores was independently associated with increased energy intake, however only the Urban pattern was also (positively) associated with increased BMI (Figure 4, Table 9).

## 4. Discussion

The present study found a high prevalence of key dietary nutrient inadequacies in a nationwide sample of Mongolian adults in summer and winter. Prior analysis of data collected from the present study population, and the Fifth National Nutrition Survey (FNNS), indicate an extremely high prevalence of biochemical vitamin D deficiency throughout the Mongolian population, especially in winter [7,9]. The present study also found a high prevalence of dietary vitamin A inadequacy, and among women, a moderate prevalence of iron inadequacy. By contrast, we found little evidence of biochemical iron and vitamin A deficiency; in the FNNS, these deficiencies were also rare, while iron deficiency was moderately prevalent among pregnant women [7]. Retinol binding protein, while useful for assessing clinical vitamin A deficiency, is subject to homeostatic regulation that renders it a less sensitive measure of subclinical deficiency (which may be widespread in Mongolia, given the high prevalence of dietary vitamin A inadequacy observed in the present study). There is also evidence that biochemical deficiencies disproportionately affect young children and pregnant women in Mongolia [7,28,34]. 

Other nutrients of concern identified in our study included fiber, calcium, magnesium, thiamin, folate, and vitamins B6, C, D, and E, while inadequacies of protein, zinc, riboflavin, niacin, and vitamin B12 were uncommon. With the exception of calcium, these findings can generally be attributed to the population’s low consumption of nutrient-dense vegetal foods (including fruits, non-tuberous vegetables, and whole grains) and high consumption of animal-source foods. Low calcium intake was more common in urban areas and in winter given lower urban and winter consumption of milk and dairy products. Inadequate intake of calcium, vitamin D, and magnesium may jointly contribute to the relatively high prevalence of osteoporosis in Mongolian adults [35], and low intake of fiber has been linked to higher risk of heart disease, type 2 diabetes, and metabolic syndrome in large prospective cohort studies [36,37,38]. Implications of observed deficits of folate and other nutrients are discussed in previous studies by our group [16], in which we have also suggested evidence-based interventions to reduce the prevalence of micronutrient deficiency in the Mongolian population, including industrial food fortification and micronutrient supplementation [9,16,39]. As micronutrients work together to allow healthy body functioning and prevent disease, addressing multiple inadequacies (particularly through large-scale diet modification in the long term) may reap multiplicative benefits, and ensuring adequate micronutrient intake should remain integral to Mongolia’s national health policy.

It is notable that fat contributed an average of 38% of caloric energy in this survey. Although we lacked data on the breakdown of fatty acid intake, as the majority of fat consumed originated from meat and dairy products, it can be assumed that saturated fat contributed a majority of fat consumed; this is supported by results from the Global Dietary Database [40]. Omega-3 fatty acid intake was also likely very low in the study population, given minimal observed intake of fish, nuts, and seeds, and the relatively low alpha-linolenic content of the dominant liquid oil consumed (sunflower seed oil) [41]. Low intake of unsaturated fats has been associated with a more deleterious cholesterol profile, insulin resistance, and higher blood pressure [42,43,44,45]; this, and a combination with a constellation of other dietary risks for chronic disease observed in our study (including high intake of refined grains, red meat, and high intake of sugary drinks in urban areas) are likely important contributors to the increasing burden of cardiovascular disease and type 2 diabetes in Mongolia [6,7,8].

In the present study, three dietary patterns explained 41% variation in consumption of major food groups, and while all were associated with increased energy intake, only the “Urban” pattern was independently associated with increased body mass index. In a prior study, Dugee and colleagues derived three exploratory diet patterns using semiquantitative food frequency questionnaire data collected from 420 healthy men and women aged >25 years from urban and rural areas of Ulaanbaatar and Khuvsgul in August 2005 [46], providing an interesting comparison with Mongolian adults in the present study in 2012–2016. Between 2005 and 2012–2016, pattern factor loadings and the variation in food intake explained by each were closely comparable, except that the 2012–2016 “Transitional” pattern contains high factor loadings for both sweets and SSBs not present in the comparable 2005 “Healthy” pattern. Likely in part due to this difference, only the 2005 “Healthy” pattern, and not its 2012–2016 “Transitional” counterpart, was associated with reduced odds of obesity in energy-adjusted analysis. Both the 2005 study’s “Transitional” pattern and its 2012–2016 “Urban” counterpart were associated with obesity in both unadjusted and energy-adjusted analyses, while the 2005 “Traditional” pattern and its 2012–2016 “Nomadic” counterpart were not, suggesting that diets in urban Mongolia have remained consistently obesogenic, while that of rural areas are not. Furthermore, our finding that adherence to the 2012–2016 “Urban” and “Nomadic” patterns decreases and increases with age, respectively, in both urban and rural areas suggests the possibility that the nation will continue to converge toward an increasingly obesogenic diet in the future.

A key finding of this study was an upward trend in BMI by age among urban Mongolian adults, but not rural ones, despite similar age trends in energy intake. One probable reason for this difference is an intensely physically-active lifestyle that many nomadic Mongolians continue to engage in even into older age, while lifestyles in urban areas have become increasingly sedentary [8]. Another likely reason is higher adherence in urban areas to the Urban diet pattern, which was positively associated with BMI after adjustment for age and energy intake. It is known that increased consumption of specific foods typically associated with industrial or western diet patterns globally (such as refined grain products and sugar-sweetened beverages, factor loadings of which were highest for the Urban pattern) are linked to long-term weight gain independent of increases in energy intake [47].

Comparable to results of the FNNS, the present study observed a low prevalence of underweight and a high prevalence of overweight and obesity among urban and rural men and non-pregnant women [7]. Nationally, from 2005 to 2013 alone, the prevalence of child stunting decreased from 27.5 to 10.8% [1], while that of overweight increased more than all but three countries globally [48]. In addition, analysis of Multiple Indicator Cluster Survey (MICS) data from 2005, 2009, and 2013 has shown consistent increases in population mean body mass index, waist circumference, percent body fat, blood pressure, and cholesterol in the adult population [8], with 61.9% and 27.5% of Mongolian adults having elevated levels of serum cholesterol and blood pressure, respectively, or on blood pressure medication. Concurrent to trends in metabolic risks, 2005 to 2013 saw a 11.4% increase in per capita caloric energy in the Mongolian Food Supply (the 15th largest increase among 175 countries during that period) and an increase in the prevalence of low physical activity (<600 MET-minutes/week) from 7.4 to 22.3% [2,8], trends that are in turn plausibly related to tremendous economic and demographic changes that have swept Mongolia as a result of the country’s recent mining boom. Since the year 2000, the fraction of Mongolians living in urban areas increased from 57.1 to 68.4%, national gross domestic production by a factor of 11.4, and total imports value by a factor of 9.6 [1]. While rapid economic growth and foreign investment, globalization of the food market, and shifts from a traditionally nomadic lifestyle to one of urban sedentism are implicated in Mongolia’s waning burden of severe undernutrition, these trends have also contributed toward the proliferation of noncommunicable diseases and have done so since the country’s tumultuous adoption of free-market reforms after 1990 [49].

Nonetheless, there is some evidence that the overall nutritional quality of the Mongolian food supply has improved from 1990 to 2010 in terms of both increased per capita supply of healthy foods and nutrients, and decreased supply of unhealthy ones (more so than any other country) [50]. Shifts in specific dietary components during this period may have also predisposed Mongolians to weight loss and other positive metabolic changes independent of changes in caloric intake [47]; these include a 69.4 to 52.6 g/day decrease in intake of sugar-sweetened beverages and a 12.9 to 25.9 g/day increase in whole grains [40]. Despite these improvements, the annual reduction in age-standardized cardiovascular, type 2 diabetes, and cancer mortality attributable to dietary imbalances from 1990 to 2010 has been minimal (−0.5%, which is close to the median change observed in countries globally), while the magnitude of reductions in under-5 DALYs attributable to wasting (−8.9%), stunting (−10.6%), protein-energy malnutrition (−11.6%), and deficiencies of iron (−7.8%), zinc (−11.0%), and other nutrients (−11.5%) during this period has each been within the top 15% of countries [6]. This is in part attributable to the fact that national indicators of undernutrition respond relatively sensitively to increased food and nutrient availability, while NCDs do not on account of their longer longer latency periods. It can therefore be expected that ongoing improvements in the Mongolian food supply will have a more positive impact on NCD risk in the future, and this is borne out by current models that project a substantially increased annual reduction (−2.1%) in age-standardized diet-related mortality from 2020 to 2040 [6].

To our knowledge, this has been the most detailed and expansive study of the diet of Mongolian adults to date. An important strength of this study was the use of paired sampling in summer and winter, which ensured that seasonal differences were largely un-confounded by participant characteristics. One limitation of the study was the fact that data were collected in only two seasons, although this limitation was mitigated by sampling during the peaks of summer and winter (which represent seasonal extremes in the Mongolian food supply). A second limitation was the lack of data on physical activity, which likely plays a role in differences in clinical measurements observed between subgroups, particularly urban-rural differences. To reduce bias in associational analyses, we attempted to stratify by or control for factors that may in part reflect physical activity (such as urbanicity, energy intake, or BMI), however, residual confounding is possible. Third, due to resource limitations, each year of the study involved data collection from up to three provinces only, and in each province, collection was completed over the course of one year; it was therefore not possible to examine changes in diet or nutrition that may have occurred in diet and nutrition from 2011 to 2016 (which are possible, given rapid trends in urbanization and global market integration that have occurred in Mongolia in recent years) because such analyses would be intractably confounded by region. Fourth, despite the size and breadth of the study population (comprising 1838 person-days of summer and winter prospective dietary intake from urban and rural men and women in 8 national provinces), and the use of random sampling within each province, the study population is not intended to be nationally-representative and results should be interpreted in the context of the sampling frame. Readers of our earlier fortification modeling study [16], which includes national estimates for the baseline prevalence of dietary inadequacy of ten nutrients using data from the same population as the present study, will note moderate quantitative differences between those and the present results for thiamin and vitamin A in some subgroups. These differences may be attributable to differing methodologies used in estimating intake distributions as well as our earlier use of survey weights. The latter were necessary given the objective of that study to estimate and evaluate operational parameters of a national fortification program; for the explicit purpose of describing population thiamin and vitamin A adequacy, we thus suggest treating the estimates of both studies as realistic bounds of the true values.

## 5. Conclusions

Key dietary inadequacies and overweight are highly prevalent among Mongolian adults. Long-term strategies for addressing micronutrient deficits in Mongolia will require immense investment in agricultural infrastructure, trade and procurement policies, and demand generation designed to compensate for the country’s environmental constraints that have historically limited fruit and vegetable availability. Similarly, Mongolia’s NCD strategy requires development and calls for a multi-pronged approach, but would be cost-effective [51,52]. While contingent on efforts to increase the diversity and quality of foods supplied, one important strategy that will improve both micronutrient intake and NCD risk in Mongolia will be the revision of the country’s dietary guidelines, which are both inadequately publicized and are not well-informed by nutrition science [53]. Finally, research priorities going forward should include the compilation, dissemination, and periodic update of national food composition data as many unique local dairy products and other foods cannot be easily imputed using foreign data; development, validation, and dissemination of standardized dietary assessment instruments; implementation of a national dietary surveillance system (potentially co-opting available household and commercial data platforms [54]); and continued development and follow-up of cohorts and other study populations for the purpose of nutritional epidemiology [55].

## Figures and Tables

**Figure 1 nutrients-12-01514-f001:**
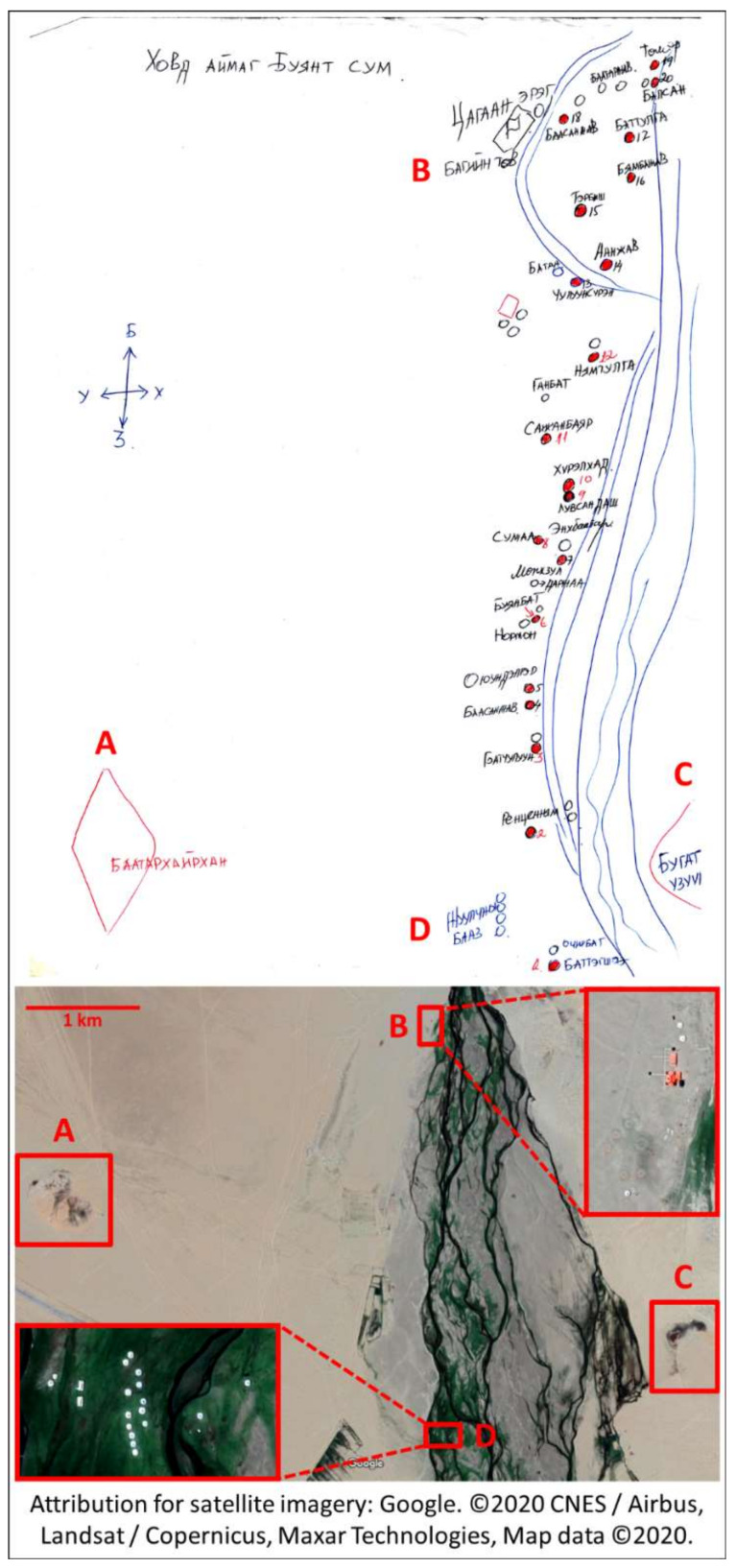
A health officer’s map of rural participant locations juxtaposed with satellite imagery. In rural areas, local health department staff are skilled at locating and tracking surrounding nomadic families, despite their frequent movements. This is exemplified by this map, quickly drafted from memory for the study team’s use by a local health office working in the rural Buyant subdistrict of Khovd province (western Mongolia). The following landmarks can be found in the map and the satellite map of the same study site: **A**: “Baatarkhairkhan” mountain; **B**: local administration center, **C**: “Deer antler” rock formation, **D**: tourist camp. Attribution for satellite imagery: Google Maps. ©2020 CNES / Airbus, Landsat / Copernicus, Maxar Technologies, Map data ©2020. https://www.google.com/maps/@47.9632414,91.593478,7154a,35y,217.12h/data=!3m1!1e3.

**Figure 2 nutrients-12-01514-f002:**
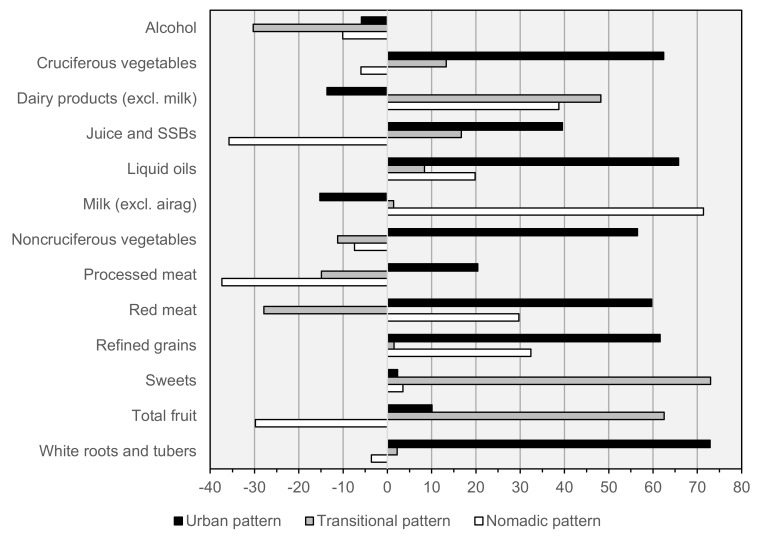
Diet pattern factor loadings. Urban, Transitional, and Nomadic diet patterns account for 20.5%, 10.9%, and 9.8% of variation in intake of pattern components, respectively (total: 41.1%). Factor loadings are generated using annualized average intakes for 334 participants (based on 1834 diet record days); used to calculate season-specific patterns scores for each participant; and seasonal pattern scores are scaled from 0–100. Airag: fermented mares’ milk, SSBs: sugar-sweetened beverages.

**Figure 3 nutrients-12-01514-f003:**
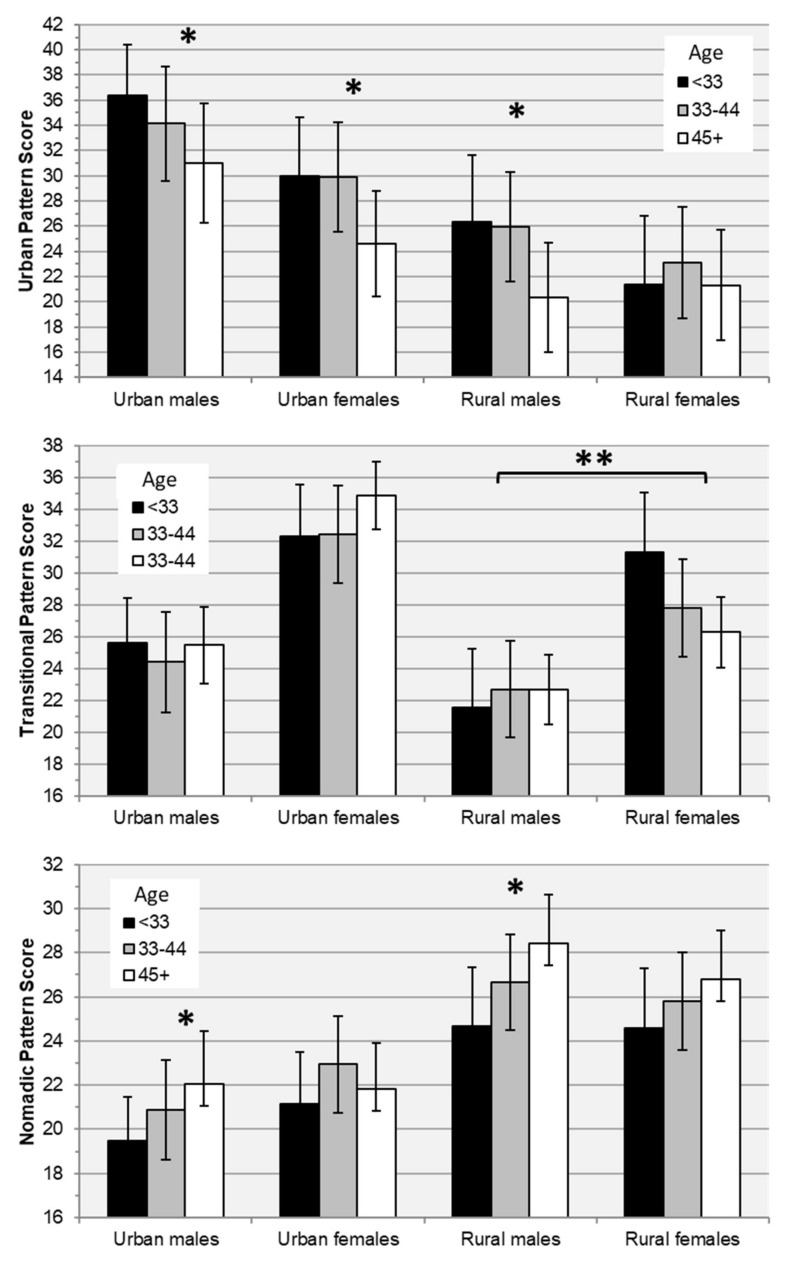
Age-trends in estimated marginal means of diet pattern scores by subgroup. Means are estimated using a regression model in which summer or winter pattern score (*n* = 635) is predicted using an age group × urbanicity-sex group interaction term adjusted for province, season, within-season mean daily energy intake, and a random intercept per person to account for within-person correlation across seasons. Error bars indicate 95% confidence intervals. * Significant linear trend with age (*p* < 0.05). ** Significant difference in age trends between rural males and females.

**Figure 4 nutrients-12-01514-f004:**
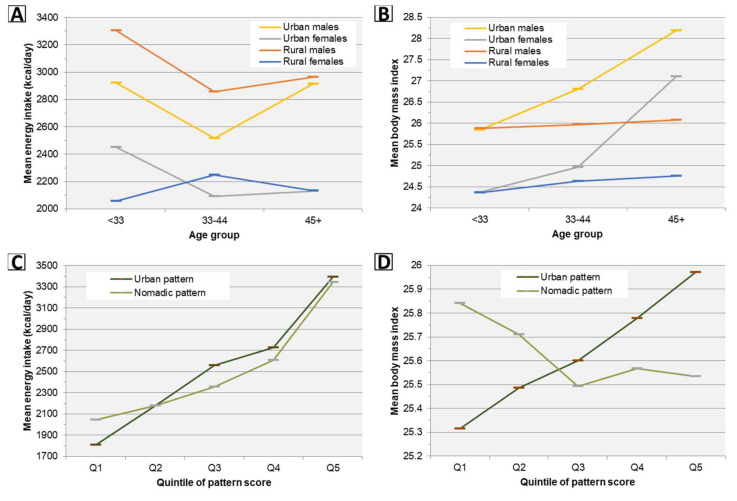
Trends in estimated marginal means of energy intake and body mass index by age and diet pattern scores across subgroups. (**A**,**B**) Means are estimated using a regression model in which summer or winter mean energy intake (*n* = 635) or BMI (*n* = 334) are predicted using an age group × urbanicity-sex group interaction term adjusted for province; energy models are additionally adjusted for season and a random intercept per person to account for within-person correlation across seasons. Within both urban males and rural males, adjusted energy intake decreases significantly (*p* < 0.05) from <33 to 33–44 years, and there are no significant differences in linear age trends between any subgroups; see Table 5 for age trends in BMI. (**C**,**D**) For each diet pattern, means are estimated using a regression model in which summer or winter mean energy intake (*n* = 635) or BMI (*n* = 334) are adjusted for quintile of pattern score, age in years, sex, urbanicity, and province; energy models are additionally adjusted season and a random intercept per person. See Table 9 for trends in energy intake and BMI across quintiles of pattern scores.

**Table 1 nutrients-12-01514-t001:** Mean usual intake (g/day) and intake density (g/2500 kcal/day) of food groups by subgroup-season and province.

	Food Group	Urban Male Summer	Urban Male Winter	Urban Female Summer	Urban Female Winter	Rural Male Summer	Rural Male Winter	Rural Female Summer	Rural Female Winter	Ulaanbaatar	Tuv	Buglan	Khuvsgul	Khovd	Omnogobi	Sukhbaatar	Dornod
		269 DR	230 DR	278 DR	246 DR	216 DR	200 DR	206 DR	193 DR	219 DR	227 DR	239 DR	231 DR	228 DR	241 DR	227 DR	226 DR
**Mean Usual Intake (g/day)**	**Fruit**	15.5	16.1	61.5	69.0	9.3	7.5	13.5	12.5	21.4	13.1	32.8	13.0	31.3	30.8	13.8	29.8
	Citrus fruits	0.0	1.0	0.0	0.1	0.0	0.0	0.0	0.0	0.0	0.0	0.0	0.0	0.0	0.0	1.1	0.0
	Deep orange fruits	3.4	0.4	3.9	1.3	2.4	0.2	0.0	0.4	1.8	2.8	1.6	1.0	0.2	0.4	0.7	5.3
	Other fruits	12.1	14.7	57.6	67.5	7.0	7.3	13.5	12.1	19.6	10.3	31.2	11.9	31.1	30.4	12.0	24.6
	**Non-Tuberous Vegetables**	64.1	61.6	61.2	46.3	50.1	41.6	26.2	28.6	64.3	55.2	42.4	46.6	35.8	49.5	41.0	54.6
	Cruciferous vegetables	14.4	10.8	13.0	11.4	13.0	7.2	3.1	5.8	15.0	9.1	12.7	7.8	8.1	9.9	9.4	9.2
	Dark green leafy vegetables	0.0	0.3	0.7	0.1	0.2	0.0	0.2	0.0	0.5	0.0	0.1	0.0	0.0	0.4	0.1	0.6
	Deep orange vegetables	0.6	0.0	0.0	0.0	0.0	0.0	0.0	0.0	0.0	0.0	0.0	0.0	0.6	0.1	0.0	0.0
	Legumes	0.4	1.9	0.2	0.5	0.0	0.0	0.0	0.0	0.0	0.6	0.0	0.5	0.0	0.0	1.9	0.0
	Other vegetables	48.8	48.5	47.3	34.2	36.8	34.5	22.8	22.8	48.7	45.5	29.6	38.3	27.2	39.1	29.6	44.7
	**Dairy**	153.4	115.8	163.6	127.2	656.5	256.7	373.9	181.9	129.0	533.4	199.4	255.9	214.6	219.1	246.3	260.6
	Milk (incl. airag)	103.3	78.5	93.6	80.4	578.7	231.3	288.8	157.6	75.4	487.7	162.7	182.2	149.1	165.9	199.3	203.6
	Dairy products	50.2	37.3	70.0	46.7	77.8	25.3	85.1	24.3	53.6	45.7	36.7	73.7	65.5	53.2	47.0	57.0
	**Eggs**	13.1	10.3	12.0	8.4	2.6	0.6	2.5	1.3	17.3	4.4	9.9	5.2	2.4	8.0	3.5	6.3
	**Juice and SSBs**	177.8	138.7	65.3	54.8	41.4	17.7	39.2	12.4	130.2	89.1	43.6	104.4	75.4	92.0	48.7	50.2
	Juice	57.2	47.2	36.8	39.8	18.1	10.7	17.7	5.0	36.2	32.6	15.0	55.5	43.1	17.9	25.8	27.0
	Sugar-sweetened beverages	120.6	91.5	28.5	14.9	23.3	7.1	21.5	7.3	94.0	56.5	28.7	48.9	32.3	74.1	22.9	23.1
	**Liquid Oils**	8.0	5.0	7.2	4.0	4.8	3.2	3.9	4.0	6.4	4.7	3.1	4.2	4.2	4.1	4.5	4.8
	**Nuts and Seeds**	1.2	0.0	2.5	0.5	1.0	0.0	1.6	0.0	0.4	1.8	0.7	0.2	0.9	2.9	1.0	0.5
	**Sweets**	13.3	12.3	27.8	34.2	10.5	12.8	15.3	16.5	21.4	15.5	16.2	17.6	18.9	25.9	19.8	17.2
	**Red and Processed Meat**	354.8	424.7	244.3	240.0	302.3	450.1	205.6	295.1	343.4	322.5	296.9	317.0	341.6	299.0	302.8	291.3
	Processed meat	15.6	14.0	4.7	3.9	2.6	4.9	6.1	0.9	23.1	6.3	2.3	9.3	4.6	2.6	2.8	7.3
	Red meat	339.3	410.7	239.6	236.1	299.8	445.2	199.5	294.2	320.2	316.2	294.6	307.8	337.0	296.4	300.1	284.0
	**Lean Meat**	5.6	5.2	5.1	1.5	0.2	13.3	1.3	6.3	4.4	6.8	0.6	1.6	1.7	15.9	4.5	1.2
	Fish	1.7	0.7	0.2	1.5	0.2	12.8	1.3	6.3	0.7	1.6	0.0	1.6	0.6	15.3	0.8	0.8
	Poultry	3.9	4.5	4.9	0.0	0.0	0.5	0.0	0.0	3.8	5.2	0.6	0.0	1.1	0.6	3.7	0.4
	**Grains**	457.9	474.1	399.9	333.5	413.5	532.8	322.1	368.7	432.3	397.2	396.5	442.2	466.9	369.0	380.6	397.1
	Refined grains	452.7	465.4	392.9	322.0	409.6	531.7	314.6	365.8	428.8	392.7	394.9	438.1	461.3	364.1	360.9	394.9
	Whole grains	5.2	8.7	7.0	11.5	3.9	1.2	7.4	2.9	3.4	4.5	1.6	4.0	5.5	4.9	19.8	2.2
	**Deep Orange Roots and Tubers**	0.0	0.0	0.0	0.0	0.0	0.0	0.0	0.0	0.0	0.0	0.0	0.0	0.0	0.0	0.0	0.0
	**White Roots and Tubers**	77.6	75.2	49.0	56.9	60.3	52.4	41.2	36.0	82.6	61.2	53.7	50.9	49.7	53.4	49.1	59.0
**Habitual Usual Density (g/2500 kcal/day)**	**Fruit**	17.4	14.1	60.6	53.1	9.4	7.7	13.8	12.9	22.7	12.1	45.4	14.9	28.4	31.5	15.7	33.2
	Citrus fruits	0.0	0.9	0.0	0.1	0.0	0.0	0.0	0.0	0.0	0.0	0.0	0.0	0.0	0.0	1.2	0.0
	Deep orange fruits	3.0	0.3	4.6	1.3	2.0	0.2	0.0	0.5	1.7	2.4	1.7	1.0	0.1	0.6	0.9	5.8
	Other fruits	14.4	12.8	56.0	51.6	7.4	7.5	13.8	12.4	21.0	9.7	43.7	13.9	28.3	30.9	13.6	27.4
	**Non-tuberous vegetables**	60.6	56.2	75.0	61.8	43.3	32.9	35.6	32.4	63.5	56.3	44.3	46.4	32.9	58.8	43.6	61.2
	Cruciferous vegetables	14.1	10.1	12.9	13.8	9.1	6.6	6.8	6.4	15.4	8.6	13.5	7.8	7.9	10.5	9.9	9.8
	Dark green leafy vegetables	0.0	0.3	1.0	0.2	0.3	0.0	0.3	0.0	0.5	0.0	0.1	0.0	0.0	0.6	0.1	0.7
	Deep orange vegetables	0.5	0.0	0.0	0.0	0.0	0.0	0.0	0.0	0.0	0.0	0.0	0.0	0.6	0.1	0.0	0.0
	Legumes	0.3	2.0	0.2	0.7	0.0	0.0	0.0	0.0	0.0	0.6	0.0	0.6	0.0	0.0	2.2	0.0
	Other vegetables	45.7	43.8	61.0	47.0	33.9	26.3	28.4	26.0	47.5	47.1	30.6	38.0	24.5	47.5	31.4	50.7
	**Dairy**	175.1	100.5	205.9	156.8	606.6	210.7	461.2	213.6	138.5	497.5	219.5	231.5	202.7	243.9	245.7	290.1
	Milk (incl. airag)	130.2	79.8	126.4	101.9	533.8	191.2	355.3	187.5	76.9	453.0	175.8	157.8	144.0	186.8	205.9	226.3
	Dairy products	44.9	20.7	79.5	54.9	72.8	19.4	105.9	26.0	61.6	44.5	43.7	73.7	58.7	57.1	39.8	63.8
	**Eggs**	12.4	10.4	13.8	12.8	2.5	0.8	2.3	1.5	15.0	4.3	12.5	5.1	3.3	9.0	4.8	7.4
	**Juice and SSBs**	188.1	114.4	84.9	68.3	35.1	12.2	41.0	13.8	121.1	78.5	51.4	106.5	59.1	94.0	57.8	53.9
	Juice	42.7	48.9	53.6	49.7	11.8	7.1	13.6	5.7	39.5	29.6	18.8	58.4	39.7	19.3	32.0	26.2
	Sugar-sweetened beverages	145.4	65.6	31.3	18.6	23.3	5.2	27.4	8.1	81.6	48.9	32.6	48.2	19.4	74.8	25.9	27.7
	**Liquid Oils**	6.0	4.5	4.7	3.8	4.4	3.9	3.9	3.9	6.2	4.3	3.5	3.9	3.8	4.2	4.5	5.0
	**Nuts and Seeds**	0.7	0.0	2.5	0.4	0.6	0.0	1.6	0.0	0.2	1.3	0.6	0.2	0.8	2.7	0.9	0.6
	**Sweets**	15.8	13.8	31.5	32.3	9.6	8.6	20.5	19.7	22.5	16.1	19.7	17.8	16.9	24.3	20.1	18.9
	**Red and Processed Meat**	324.4	376.1	260.4	303.0	286.3	359.7	250.2	342.3	310.7	296.4	321.0	297.4	295.2	317.6	315.7	295.2
	Processed meat	12.7	12.7	5.2	5.9	2.8	2.9	8.6	1.0	18.9	6.0	2.4	8.7	4.1	3.4	3.4	8.5
	Red meat	311.7	363.4	255.2	297.1	283.5	356.9	241.5	341.3	291.8	290.4	318.6	288.6	291.1	314.2	312.3	286.7
	**Lean Meat**	3.7	6.0	5.2	1.9	0.3	9.8	1.6	5.9	4.7	6.4	0.4	1.8	2.4	14.7	4.7	1.4
	Fish	1.7	0.7	0.4	1.9	0.3	9.4	1.6	5.9	0.7	1.8	0.0	1.8	1.6	14.0	1.1	0.9
	Poultry	2.1	5.3	4.7	0.0	0.0	0.3	0.0	0.0	4.0	4.6	0.4	0.0	0.9	0.6	3.7	0.5
	**Grains**	400.3	421.2	425.0	407.4	372.1	407.3	381.0	419.7	412.0	376.0	404.9	448.3	410.2	379.5	400.7	408.5
	Refined grains	395.2	417.5	415.8	398.1	368.0	406.2	373.7	415.8	408.8	371.9	402.7	444.4	405.5	375.3	380.1	406.4
	Whole grains	5.1	3.7	9.3	9.3	4.1	1.1	7.3	3.8	3.2	4.1	2.3	3.9	4.6	4.2	20.6	2.0
	**Deep Orange Roots and Tubers**	0.0	0.0	0.0	0.0	0.0	0.0	0.0	0.0	0.0	0.0	0.0	0.0	0.0	0.0	0.0	0.0
	**White Roots and Tubers**	74.0	67.4	55.8	70.7	55.1	43.3	50.2	40.6	78.2	56.2	58.2	49.6	46.7	56.7	52.7	61.7

The total number of diet records (DR) available for analysis are listed below each column heading. Depending on data availability, usual intakes within subgroup-seasons are either estimated using habitual or episodic intake models analyzing up to 3 days per person per season, or calculated by first computing each person’s average daily intake in each season and then averaging these within-person seasonal means across persons in each subgroup-season (intake densities are only calculated using this sequential averaging procedure). Usual intakes and intake densities within provinces are calculated by first averaging each person’s summer and winter daily means and then averaging these within-person annual means across persons in each province. To aid interpretation, intake densities are expressed on a per 2500 kcal/day basis, which is approximately equal to the mean daily energy intake in the study population. Intake models are estimated using SPADE [31]. Statistics are weighted by weekday of each record day. Airag: fermented mares’ milk, SSBs: sugar-sweetened beverages.

**Table 2 nutrients-12-01514-t002:** Mean usual nutrient intake (per day) and prevalence of nutrient inadequacy (% < estimated average requirement) by subgroup-season and province.

	Nutrient	Urban Male Summer	Urban Male Winter	Urban Female Summer	Urban Female Winter	Rural Male Summer	Rural Male Winter	Rural Female Summer	Rural Female Winter	Ulaanbaatar	Tuv	Buglan	Khuvsgul	Khovd	Omnogobi	Sukhbaatar	Dornod
		269 DR	230 DR	278 DR	246 DR	216 DR	200 DR	206 DR	193 DR	219 DR	227 DR	239 DR	231 DR	228 DR	241 DR	227 DR	226 DR
**Mean Usual Intake (per day)**	Energy (kcal)	2972	2897	2339	2069	2752	3266	2113	2214	2603	2664	2359	2634	2857	2407	2441	2478
	Protein (g)	160.96	161.56	113.10	101.98	145.41	171.62	103.99	115.10	136.30	149.76	119.60	128.01	139.26	130.71	132.66	124.68
	per kg	2.1	2.1	1.8	1.6	1.9	2.3	1.7	1.9	2.0	2.1	1.5	1.8	2.1	2.0	1.9	1.8
	% total energy	21.3	22.4	19.1	20.1	21.0	21.5	19.9	21.4	20.9	21.6	20.4	19.3	19.5	21.8	21.4	20.4
	Available carbohydrates (g)	284.73	267.29	240.22	211.99	256.70	295.44	208.60	213.22	253.82	250.57	227.59	258.22	267.06	230.48	237.75	245.51
	% total energy	40.8	38.8	43.8	42.4	37.8	37.4	40.2	39.5	41.2	38.9	39.8	41.8	38.9	40.0	40.9	41.6
	Total fat (g)	121.15	120.32	97.58	85.47	117.75	144.80	92.97	96.28	109.79	111.51	103.89	111.30	128.98	99.32	98.11	102.98
	% total energy	36.0	37.4	36.6	37.0	37.5	39.8	39.2	38.9	37.6	36.1	38.9	37.1	41.0	36.9	36.5	36.9
	Alcohol, (g)	6.86	7.00	1.66	0.94	14.98	5.15	2.66	0.37	1.15	15.91	2.68	7.38	3.13	6.02	4.65	4.97
	% total energy	2.0	1.4	0.5	0.4	3.7	1.2	0.7	0.2	0.3	3.4	0.8	1.8	0.6	1.3	1.2	1.0
	Fiber (g)	11.7	10.9	11.0	9.1	9.3	10.7	7.2	7.9	11.1	9.7	8.8	9.7	10.6	8.9	9.1	10.3
	Phytosterols (mg)	455	431	386	333	416	500	328	353	409	397	384	365	440	388	386	408
	Calcium (mg)	663	518	594	461	1361	730	852	515	541	1092	573	709	626	640	683	757
	Phytate:Ca molar ratio	0.1	0.1	0.1	0.1	0.0	0.1	0.0	0.1	0.1	0.0	0.1	0.0	0.1	0.1	0.0	0.0
	Copper (mg)	1.933	1.932	1.731	1.322	2.642	2.078	2.101	1.405	1.723	1.777	1.603	1.872	2.060	1.860	2.013	2.056
	Iron (mg)	20.61	20.86	15.49	13.48	18.98	23.38	14.52	16.18	17.77	18.45	15.02	18.84	19.84	16.99	17.72	17.79
	Phytate:Fe molar ratio	2.1	1.9	2.5	2.3	2.3	2.0	2.4	2.0	2.3	2.1	2.4	1.9	2.1	2.5	2.3	2.3
	Magnesium (mg)	345	367	277	251	351	389	244	277	316	367	284	323	329	274	302	309
	Manganese (mg)	4.170	4.054	3.266	2.663	3.166	4.056	2.411	2.966	3.584	3.345	3.163	3.526	3.818	2.967	3.054	3.200
	Phosphorus (mg)	1501	1519	1145	1042	1649	1864	1251	1260	1303	1497	1203	1376	1591	1355	1363	1392
	Potassium (mg)	3234	3478	2473	2335	2996	3538	2166	2457	2892	3153	2532	2789	2948	2670	2788	2724
	Zinc (mg)	22.32	24.79	16.95	15.23	20.50	25.37	14.61	16.56	21.05	19.99	18.13	19.70	21.16	19.60	18.43	18.13
	Phytate:Zn molar ratio	2.1	2.0	2.5	2.5	2.3	2.3	2.5	2.4	2.2	2.3	2.3	2.1	2.4	2.4	2.4	2.6
	Vitamin A (ug retinol eq.)	577	559	628	413	1008	490	808	308	526	514	507	653	611	551	632	680
	% retinol	51.8	54.8	63.0	59.4	73.6	67.3	80.6	68.5	55.4	61.5	59.1	64.4	66.7	67.5	70.1	69.7
	Thiamin (mg)	1.259	1.112	1.027	0.810	1.414	1.156	0.970	0.811	1.096	1.203	0.839	1.230	1.125	0.970	0.980	1.203
	Riboflavin (mg)	2.178	2.040	1.801	1.397	2.672	2.358	2.225	1.560	1.827	1.846	1.564	2.173	2.261	2.010	2.157	2.197
	Niacin (mg)	27.406	27.237	19.954	16.250	24.086	27.311	17.022	17.575	23.778	22.445	18.710	23.653	23.452	22.323	21.217	21.622
	Pantothenic acid (mg)	7.569	7.480	5.605	4.998	7.638	8.093	5.812	5.641	6.568	6.637	5.583	6.472	7.318	6.346	7.020	6.660
	Vitamin B6 (mg)	1.332	1.480	0.920	0.917	1.018	1.394	0.786	0.980	1.156	1.138	0.939	1.195	1.050	1.026	1.103	1.119
	Folate (ug DFE)	166	163	143	119	170	191	128	135	142	149	133	162	219	140	127	138
	Vitamin B12 (ug)	13.18	12.22	8.31	6.86	17.64	12.96	15.15	8.79	10.81	10.70	9.60	10.27	13.28	12.21	13.87	11.66
	Vitamin C (mg)	35.3	42.2	33.2	38.0	25.6	22.7	17.6	13.4	38.3	28.0	21.0	31.7	31.8	26.5	21.8	24.4
	Vitamin D (IU)	39	31	34	27	45	43	36	25	40	37	31	35	27	43	36	36
	Vitamin E (mg)	8.31	8.62	6.88	5.60	6.86	9.25	5.49	6.73	7.92	7.36	6.77	7.08	8.94	6.65	6.02	6.98
**Prevalence of Inadequacy (% < EAR)**	Protein (per kg)	0.0	0.2	2.6	3.0	0.1	0.0	0.6	0.2	0.0	0.0	0.4	0.0	0.1	0.2	0.3	0.3
	Calcium	74.1	94.6	85.4	95.3	31.8	66.7	55.5	88.9	93.7	40.8	88.7	71.5	95.2	78.5	72.6	65.8
	Copper	0.1	0.0	0.3	2.8	0.1	0.0	0.1	0.8	0.1	0.0	0.2	0.0	0.0	0.0	0.0	0.0
	Iron	0.0	0.0	19.0	21.4	0.1	0.0	15.7	14.1	7.5	7.4	7.5	6.6	7.3	8.9	7.4	6.9
	Magnesium	51.6	47.6	48.5	60.1	55.5	33.4	64.6	47.9	48.1	33.6	64.6	42.3	43.6	66.2	54.8	54.5
	Phosphorus	0.0	0.1	5.0	9.6	0.4	0.0	0.2	1.0	1.1	0.1	1.7	0.1	0.0	0.7	0.6	0.6
	Zinc	0.1	0.2	1.7	0.2	0.4	0.1	0.2	0.8	0.6	0.3	1.3	0.0	0.2	0.4	0.0	2.1
	Vitamin A	65.8	68.9	48.5	73.5	41.8	75.9	20.4	89.7	65.5	66.5	66.9	41.3	45.6	61.0	51.0	29.1
	Thiamin	28.9	43.8	46.5	67.0	27.1	35.8	44.3	67.5	45.0	26.5	69.7	28.0	20.3	52.0	49.3	35.1
	Riboflavin	2.5	0.7	3.8	13.5	3.7	0.7	0.3	4.1	4.7	0.0	7.9	0.5	0.0	1.3	1.6	2.2
	Niacin	0.2	0.6	8.1	12.6	2.5	0.3	5.5	8.6	5.0	1.9	11.1	0.7	1.5	1.7	0.2	8.4
	Vitamin B6	35.7	27.9	74.1	77.4	67.5	23.5	90.7	72.1	53.1	54.8	86.8	52.1	62.8	65.3	59.8	58.1
	Folate	98.7	97.9	96.4	98.9	95.8	93.0	99.9	98.3	99.3	99.5	99.7	98.1	89.6	99.2	100.0	98.6
	Vitamin B12	0.0	0.0	0.0	0.0	0.0	0.0	0.0	0.0	0.0	0.0	0.0	0.0	0.0	0.0	0.0	0.0
	Vitamin C	98.9	98.7	96.1	87.4	99.7	100.0	100.0	100.0	100.0	98.0	100.0	99.0	98.1	98.3	100.0	99.9
	Vitamin D	100.0	100.0	100.0	100.0	100.0	100.0	100.0	100.0	100.0	100.0	100.0	100.0	100.0	100.0	100.0	100.0
	Vitamin E	95.3	89.0	94.1	98.2	95.7	88.7	100.0	93.9	95.7	97.6	97.8	98.1	90.9	96.6	99.8	94.6

The total number of diet records (DR) available for analysis are listed below each column heading. Within subgroup-seasons and provinces, usual intakes of all nutrients (except alcohol) and nutrient adequacy prevalence are estimated using habitual intake models analyzing up to 3 or 6 record days per person, respectively, while nutrient proportions and ratios are calculated using the sequential averaging procedures described in the footnote to Table 1. Depending on data availability, usual alcohol intake is estimated either using episodic intake models or calculated by sequential averaging. Estimated average requirements are drawn from the Dietary Reference Intakes of the National Academy of Medicine [33]. Intake models are estimated using SPADE [31]. Statistics are weighted by weekday of each record day. Shading is proportional to nutrient inadequacy prevalence (white: 0.0%, red: 100.0%).

**Table 3 nutrients-12-01514-t003:** Mean percentage of dietary energy contributed by consumed dishes by subgroup-season and province.

Food Group	Urban Male Summer	Urban Male Winter	Urban Female Summer	Urban Female Winter	Rural Male Summer	Rural Male Winter	Rural Female Summer	Rural Female Winter
	269 DR	230 DR	278 DR	246 DR	216 DR	200 DR	206 DR	193 DR
Airag	0.9	0.1	0.3	0.0	4.5	0.2	1.5	0.3
Beer, wine, spirits	2.8	2.3	0.8	0.7	4.4	2.6	0.4	0.2
Biscuits, cookies, doughnuts	8.1	10.1	13.2	16.4	12.3	20.6	19.1	20.7
Bread with or without toppings	10.5	8.2	11.8	9.5	11.2	4.5	11.5	4.7
Dairy products (excl. milk), eggs	3.8	2.3	6.3	5.7	4.9	3.1	7.7	4.4
Dumplings (excl. in soup)	11.7	19.3	10.7	15.9	7.6	20.5	5.7	19.2
Juice, SSBs	3.4	2.1	1.9	1.5	0.7	0.3	0.8	0.2
Milk (excl. airag), milk tea	4.1	4.5	5.6	5.4	12.0	8.5	13.6	9.9
Miscellaneous foods	1.6	1.3	1.2	1.7	0.8	0.8	0.7	1.2
Nutrient-dense snacks	0.6	0.6	3.2	1.9	0.3	0.3	0.7	0.4
Other meat dishes	8.6	12.1	5.9	7.5	9.5	10.7	8.8	9.1
Soups	16.6	14.0	13.5	11.2	13.0	12.5	11.8	12.8
Stir fries	24.8	21.0	20.0	17.8	18.2	14.0	15.2	14.0
Sweets, ice cream	1.2	1.2	3.8	3.4	0.6	1.0	2.2	2.5
Tea, coffee, water	1.1	0.9	1.7	1.5	0.1	0.6	0.1	0.5
	**Ulaanbaatar**	**Tuv**	**Bulgan**	**Khuvsgul**	**Khovd**	**Omnogobi**	**Sukhbaatar**	**Dornod**
	**219 DR**	**227 DR**	**239 DR**	**231 DR**	**228 DR**	**241 DR**	**227 DR**	**226 DR**
Airag	0.1	4.7	1.6	0.3	0.0	1.0	0.0	0.1
Beer, wine, spirits	0.6	4.0	0.7	2.7	1.1	1.3	2.3	1.9
Biscuits, cookies, doughnuts	9.2	12.3	19.1	12.5	23.2	16.2	11.6	12.4
Bread with or without toppings	11.2	8.1	4.9	12.8	7.1	6.8	10.4	13.2
Dairy products (excl. milk), eggs	5.6	4.9	4.5	4.3	6.0	5.7	2.7	4.8
Dumplings (excl. in soup)	10.2	13.5	15.8	15.5	12.7	12.5	14.2	11.1
Juice, SSBs	2.3	1.4	1.0	2.1	1.4	1.9	1.1	1.1
Milk (excl. airag), milk tea	4.1	6.8	4.9	6.0	6.7	8.5	13.3	11.0
Miscellaneous foods	1.7	1.4	0.7	1.6	0.7	0.7	1.0	1.6
Nutrient-dense snacks	0.7	0.9	1.5	0.8	0.9	1.7	1.2	1.2
Other meat dishes	9.1	7.7	8.0	6.9	9.3	12.4	8.7	8.5
Soups	16.0	13.6	14.2	15.3	12.1	10.6	14.6	11.7
Stir fries	25.7	18.0	19.3	16.2	16.0	17.6	15.2	19.3
Sweets, ice cream	1.7	1.9	2.3	1.9	1.9	2.4	2.2	1.7
Tea, coffee, water	1.9	0.6	1.3	1.1	1.1	0.6	1.5	0.4

The total number of diet records (DR) available for analysis are listed in each column heading. Within subgroup-seasons and provinces, means are calculated using the sequential averaging procedures described in the footnote to Table 1. Shading is proportional to mean percentage contribution (white: 0.0%, blue: 25.7%). Airag: fermented mares’ milk, SSBs: sugar-sweetened beverages. See Appendix A for a brief description of each food group.

**Table 4 nutrients-12-01514-t004:** Mean body mass index and serum biomarker concentrations by subgroup, subgroup-season, and province.

Measurement	Urban Male (93 Participants)				Urban Female (96 Participants)				Rural Male (74 Participants)				Rural Female (71 Participants)			
	Mean or *n*		SE or %		Mean or *n*		SE or %		Mean or *n*		SE or %		Mean or *n*		SE or %	
Mean BMI (kg/m^3) *	26.9		0.4		25.5		0.4		26.0		0.5		24.6		0.5	
Underweight (<18.5)	1		1.1%		3		3.1%		0		0.0%		1		1.4%	
Normal (18.5–25)	29		31.2%		38		39.6%		35		47.3%		39		54.9%	
Overweight (25–30)	47		50.5%		45		46.9%		29		39.2%		28		39.4%	
Obese (30+)	16		17.2%		10		10.4%		10		13.5%		3		4.2%	
	**Urban Male Summer**		**Urban Male Winter**		**Urban Female Summer**		**Urban Female winter**		**Rural Male Summer**		**Rural Male Winter**		**Rural Female Summer**		**Rural Female Winter**	
	**90 Measurements**		**83 Measurements**		**93 Measurements**		**87 Measurements**		**72 Measurements**		**72 Measurements**		**70 Measurements**		**68 Measurements**	
	**Mean or *n***	**SE or %**	**Mean or *n***	**SE or %**	**Mean or *n***	**SE or %**	**Mean or *n***	**SE or %**	**Mean or *n***	**SE or %**	**Mean or *n***	**SE or %**	**Mean or *n***	**SE or %**	**Mean or *n***	**SE or %**
Mean ferritin (ug/L)	162.1	4.7	**157.7**	4.8	93.5	4.7	**90.8**	5.5	153.2	5.4	139.7	5.5	106.3	5.4	**106.7**	5.5
Iron deficient **	0	0.0%	0	0.0%	2	2.2%	5	5.9%	0	0.0%	0	0.0%	4	5.9%	1	1.6%
Iron overload **	0	0.0%	0	0.0%	1	1.1%	1	1.2%	0	0.0%	0	0.0%	0	0.0%	2	3.2%
Mean sTfR (mg/L)	4.79	0.22	4.69	0.23	4.95	0.22	5.22	0.26	4.77	0.25	4.78	0.26	4.95	0.26	4.81	0.26
Mean iron stores (mg/kg)	11.31	0.24	11.25	0.25	8.78	0.24	**8.62**	0.29	11.07	0.28	10.68	0.28	9.28	0.28	**9.53**	0.29
Mean RBP (umol/L)	2.35	0.06	**2.13**	0.06	1.65	0.06	1.59	0.07	2.24	0.07	**1.93**	0.07	1.79	0.07	1.50	0.07
Vitamin A deficient **	0	0.0%	0	0.0%	1	1.1%	0	0.0%	0	0.0%	1	1.5%	0	0.0%	0	0.0%
Mean CRP (mg/L)	3.60	0.51	1.62	0.55	1.92	0.51	1.52	0.62	3.71	0.59	1.79	0.60	2.80	0.59	2.64	0.62
Elevated (>5)	14	15.6%	4	5.1%	8	8.7%	6	7.1%	12	17.4%	6	9.2%	8	11.8%	7	11.3%
Mean AGP (g/L)	0.67	0.03	0.61	0.03	**0.57**	0.03	0.56	0.03	0.72	0.03	0.65	0.03	**0.69**	0.03	0.59	0.03
Elevated (>1)	9	10.0%	3	3.8%	3	3.3%	1	1.2%	10	14.5%	6	9.2%	10	14.7%	1	1.6%
	**Ulaanbaatar**		**Tuv**		**Bulgan**		**Khuvsgul**		**Khovd**		**Omnogobi**		**Sukhbaatar**		**Dornod**	
	**77 Measurements**		**80 Measurements**		**81 Measurements**		**78 Measurements**		**77 Measurements**		**86 Measurements**		**79 Measurements**		**77 Measurements**	
	**Mean or *n***	**SE or %**	**Mean or *n***	**SE or %**	**Mean or *n***	**SE or %**	**Mean or *n***	**SE or %**	**Mean or *n***	**SE or %**	**Mean or *n***	**SE or %**	**Mean or *n***	**SE or %**	**Mean or *n***	**SE or %**
Mean BMI (kg/m^3)	24.6	0.6	25.7	0.6	27.8	0.6	25.8	0.6	25.2	0.6	25.0	0.6	26.3	0.6	25.6	0.6
Underweight (<18.5)	1	2.5%	0	0.0%	0	0.0%	0	0.0%	0	0.0%	2	4.3%	1	2.5%	1	2.5%
Normal (18.5–25)	22	55.0%	19	47.5%	9	19.1%	15	37.5%	18	43.9%	26	56.5%	14	35.0%	18	45.0%
Overweight (25–30)	16	40.0%	16	40.0%	26	55.3%	19	47.5%	21	51.2%	13	28.3%	20	50.0%	18	45.0%
Obese (30+)	1	2.5%	5	12.5%	12	25.5%	6	15.0%	2	4.9%	5	10.9%	5	12.5%	3	7.5%
Mean ferritin (ug/L)	125.7	8.0	130.4	8.0	127.2	7.5	107.8	8.0	108.1	8.0	139.2	7.8	137.5	8.0	133.1	8.0
Iron deficient **	1	2.5%	0	0.0%	1	2.1%	0	0.0%	2	4.9%	3	6.5%	0	0.0%	0	0.0%
Iron overload **	1	2.5%	0	0.0%	0	0.0%	0	0.0%	0	0.0%	1	2.2%	0	0.0%	0	0.0%
Mean sTfR (mg/L)	4.96	0.32	4.98	0.32	4.89	0.30	5.37	0.32	4.69	0.32	4.92	0.31	4.72	0.32	4.49	0.32
Mean iron stores (mg/kg)	9.89	0.39	10.05	0.39	10.02	0.36	9.21	0.39	9.40	0.39	10.69	0.38	10.45	0.38	10.67	0.39
Mean RBP (umol/L)	1.82	0.09	2.04	0.08	2.08	0.08	1.83	0.09	1.86	0.08	1.86	0.08	1.98	0.08	1.77	0.08
Vitamin A deficient **	0	0.0%	0	0.0%	1	2.1%	0	0.0%	1	2.4%	3	6.5%	0	0.0%	0	0.0%
Mean CRP (mg/L)	1.63	0.60	2.62	0.58	2.49	0.57	2.49	0.59	2.29	0.59	2.55	0.59	2.33	0.57	3.02	0.58
Elevated (>5)	1	2.5%	6	15.0%	4	8.5%	5	12.5%	4	9.8%	4	8.7%	3	7.5%	5	12.5%
Mean AGP (g/L)	0.53	0.03	0.68	0.03	0.63	0.03	0.66	0.03	0.65	0.03	0.59	0.03	0.72	0.03	0.59	0.03
Elevated (>1)	1	2.5%	5	12.5%	3	6.4%	4	10.0%	2	4.9%	0	0.0%	5	12.5%	0	0.0%

Within provinces, means are calculated by first averaging each person’s summer and winter measurements then averaging these within-person annual means across persons in each province; provincial frequencies are based on within-person annual means. Significant differences in means (*p* < 0.05) based on least squares analysis are indicated for the following planned comparisons: summer vs. winter within urbanicity-sex (indicated in blue), males vs. females within urbanicity (BMI) or urbanicity-season (serum biomarkers) (green), urban vs. rural within sex (BMI only) or sex-season (bold), and each province vs. the mean of all other provinces (purple) (combinations of colors and/or bolding indicate multiple significant comparisons). * Seasonal means and differences in BMI are not assessed because height and weight were collected only in winter for the majority of participants. ** Iron and vitamin A deficiency are defined according to serum ferritin and RBP concentration cutoffs, respectively, which are differentially adjusted for inflammatory phase (normal, incubation, early convalescence, late convalescence) defined according to different combinations of elevated CRP and AGP (>5 mg/L and >1 g/L, respectively) [21]. Iron overload is defined as serum ferritin >300 ng/mL in men and >200 ng/mL in women without the presence of inflammation [22]. BMI: body mass index; sTfR: soluble transferrin receptor; RBP: retinol binding protein; CRP: C-reactive protein; AGP: alpha 1-acid glycoprotein.

**Table 5 nutrients-12-01514-t005:** Estimated marginal means of body mass index and serum biomarker concentrations by urbanicity and age.

Stratum	Age Group	BMI (kg/m^3^)		Ferritin (ug/L)		sTfR (mg/L)		Iron stores (mg/kg)		RBP (umol/L)		CRP (mg/L)		AGP (g/L)	
		Mean	SE	Mean	SE	Mean	SE	Mean	SE	Mean	SE	Mean	SE	Mean	SE
Urban	All ages	**26.4**	0.6	129.2	6.6	4.88	0.33	10.15	0.35	**1.95**	0.08	2.23	0.67	**0.61**	0.04
Rural		**24.9**	0.6	122.2	6.8	4.88	0.35	9.94	0.36	**1.84**	0.08	2.63	0.68	**0.66**	0.04
Urban Male	<33	25.8	0.5	150.5	5.7	4.88	0.29	10.93	0.30	2.12	0.07	2.50	0.56	**0.64**	0.03
	33–44	26.8	0.7	159.5	7.8	5.01	0.40	11.15	0.41	2.22	0.09	3.76	0.77	**0.62**	0.04
	45+	28.2	0.8	177.6	7.9	4.22	0.40	12.04	0.42	2.47	0.09	1.83	0.76	**0.66**	0.04
Urban Female	<33	24.4	0.6	74.8	6.9	5.61	0.35	7.49	0.37	1.61	0.08	1.86	0.69	0.59	0.04
	33–44	25.0	0.5	73.7	6.4	5.11	0.33	8.03	0.34	1.64	0.08	1.51	0.65	0.57	0.04
	45+	27.1	0.6	129.3	6.6	4.76	0.33	10.45	0.35	1.65	0.08	1.75	0.64	0.53	0.04
Rural Male	<33	25.9	0.7	131.2	9.6	4.55	0.49	10.59	0.51	2.07	0.11	2.04	0.94	**0.82**	0.05
	33–44	26.0	0.6	151.6	7.2	5.30	0.36	10.75	0.38	2.07	0.08	2.98	0.71	**0.66**	0.04
	45+	26.1	0.6	147.4	7.1	4.29	0.36	11.15	0.37	2.09	0.08	2.95	0.70	**0.64**	0.04
Rural Female	<33	24.4	0.7	79.0	9.2	4.98	0.47	7.84	0.48	1.70	0.11	2.54	0.91	0.60	0.05
	33–44	24.6	0.6	103.2	6.8	5.02	0.35	9.46	0.36	1.57	0.08	2.44	0.68	0.70	0.04
	45+	24.8	0.6	128.6	7.8	4.60	0.40	10.41	0.41	1.69	0.09	3.12	0.76	0.61	0.04

In urban vs. rural analysis, mean BMI, CRP, and AGP are adjusted for age in years, sex, province, season, and a random intercept per person to account for within-person correlation across seasons. In analysis of age, mean BMI, CRP, and AGP are estimated with a model in which the outcome in either season is predicted by an age group × urbanicity-sex group interaction term adjusted for province, season, and a random intercept per person. Within urbanicity-sex groups, underlined type indicates a significant linear trend with age (*p* < 0.05). Significant differences in trends are indicated for the following planned comparisons: urban vs. rural (or, in analysis of age, urban vs. rural within sex) (indicated in bold) and males vs. females within urbanicity (green). In both urbanicity and age analyses, mean ferritin, sTfR, body iron stores, and RBP are additionally adjusted for CRP and AGP. BMI: body mass index; sTfR: soluble transferrin receptor; RBP: retinol binding protein; CRP: C-reactive protein; AGP: alpha 1-acid glycoprotein.

**Table 6 nutrients-12-01514-t006:** Diet pattern factor loadings and mean intake (g/day) of pattern components by quintile of pattern scores.

	Component	Loading	Q1		Q2		Q3		Q4		Q5		*p*, Trend
			Mean	SE	Mean	SE	Mean	SE	Mean	SE	Mean	SE	
**Urban Pattern**	Alcohol	−5.9	8.56	1.55	4.95	1.54	4.04	1.55	3.68	1.52	5.54	1.55	**0.00**
	Cruciferous vegetables	62.4	2.9	0.8	5.6	0.8	7.5	0.8	12.2	0.8	20.4	0.8	**0.00**
	Dairy products (excl. milk)	−13.6	69.8	6.8	49.9	6.8	57.1	6.8	44.6	6.8	35.1	6.8	**0.01**
	Juice and SSBs	39.5	22.4	11.7	44.6	11.7	57.4	11.8	86.7	11.6	162.1	11.7	**0.00**
	Liquid oils	65.8	1.5	0.3	2.8	0.3	4.0	0.3	5.8	0.3	8.4	0.3	**0.00**
	Milk (excl. airag)	−15.2	168.8	17.4	170.9	17.4	147.5	17.5	111.9	17.1	93.8	17.5	**0.01**
	Noncruciferous vegetables	56.5	15.4	3.1	23.4	3.1	34.3	3.1	47.2	3.0	74.2	3.1	**0.00**
	Processed meat	20.4	0.7	2.4	5.0	2.4	5.2	2.4	7.1	2.4	17.5	2.4	**0.00**
	Red meat	59.7	178.7	12.0	232.5	12.0	315.4	12.0	348.1	11.9	441.2	12.0	**0.00**
	Refined grains	61.6	248.1	14.1	324.5	14.1	396.3	14.2	455.9	13.9	574.4	14.2	**0.00**
	Sweets	2.3	16.2	2.5	20.5	2.5	19.1	2.5	19.3	2.5	16.3	2.5	0.53
	Total fruit	10.1	17.0	4.4	17.4	4.4	27.1	4.4	29.2	4.4	25.0	4.4	**0.00**
	White roots and tubers	72.9	23.5	2.9	36.4	2.9	51.8	2.9	67.7	2.9	103.0	2.9	**0.00**
**Transitional Pattern**	Alcohol	−30.3	14.76	1.47	3.85	1.47	3.95	1.47	2.19	1.47	2.03	1.48	**0.00**
	Cruciferous vegetables	13.3	7.9	1.0	7.5	1.0	10.3	1.0	11.3	1.0	11.7	1.0	**0.00**
	Dairy products (excl. milk)	48.2	10.9	5.9	21.6	5.9	36.5	5.9	67.3	5.9	120.6	5.9	**0.00**
	Juice and SSBs	16.7	30.7	12.2	53.3	12.2	68.4	12.2	108.2	12.2	112.4	12.3	**0.00**
	Liquid oils	8.4	4.4	0.4	4.4	0.4	4.4	0.4	4.5	0.4	4.7	0.4	0.21
	Milk (excl. airag)	1.4	114.2	17.3	122.1	17.3	171.0	17.3	148.3	17.4	136.8	17.6	**0.00**
	Noncruciferous vegetables	−11.2	53.9	3.5	37.1	3.5	34.1	3.5	36.0	3.5	33.3	3.5	**0.00**
	Processed meat	−14.9	14.4	2.4	3.6	2.4	6.5	2.4	5.7	2.4	5.4	2.5	**0.02**
	Red meat	−27.9	431.5	13.2	304.6	13.2	259.2	13.2	276.5	13.2	244.3	13.2	**0.00**
	Refined grains	1.5	428.0	17.2	394.0	17.3	380.6	17.2	400.1	17.3	396.3	17.4	**0.24**
	Sweets	73.0	4.0	2.0	5.4	2.0	12.9	2.0	22.3	2.0	46.6	2.0	**0.00**
	Total fruit	62.5	2.4	3.6	6.3	3.6	10.3	3.6	21.8	3.6	74.5	3.7	**0.00**
	White roots and tubers	2.2	55.9	3.9	53.8	3.9	57.9	3.9	56.8	3.9	58.0	3.9	0.17
**Nomadic Pattern**	Alcohol	−10.1	10.46	1.54	5.22	1.52	5.16	1.52	3.89	1.52	2.04	1.56	**0.00**
	Cruciferous vegetables	−6.0	12.1	1.0	9.4	1.0	9.2	1.0	7.9	1.0	10.1	1.0	0.68
	Dairy products (excl. milk)	38.7	25.3	6.4	29.5	6.4	42.3	6.4	59.6	6.4	99.9	6.4	**0.00**
	Juice and SSBs	−35.8	186.5	11.4	77.1	11.3	52.1	11.3	30.2	11.3	27.2	11.5	**0.00**
	Liquid oils	19.8	3.5	0.4	3.4	0.4	4.1	0.4	5.3	0.4	6.2	0.4	**0.00**
	Milk (excl. airag)	71.4	56.0	15.8	81.1	15.7	108.0	15.6	154.8	15.6	295.4	16.0	**0.00**
	Noncruciferous vegetables	−7.4	42.8	3.6	46.9	3.6	34.3	3.6	32.3	3.6	38.2	3.6	0.69
	Processed meat	−37.4	24.0	2.3	4.9	2.3	3.7	2.3	1.3	2.3	1.5	2.4	**0.00**
	Red meat	29.7	224.6	13.7	259.1	13.6	310.1	13.6	323.8	13.6	397.7	13.8	**0.00**
	Refined grains	32.4	317.6	15.9	341.3	15.8	395.2	15.8	408.8	15.8	536.7	16.1	**0.00**
	Sweets	3.5	17.8	2.5	23.1	2.5	14.4	2.5	15.5	2.5	20.8	2.5	0.08
	Total fruit	−29.8	49.3	4.2	29.5	4.2	19.9	4.2	8.6	4.2	8.1	4.2	**0.00**
	White roots and tubers	−3.6	58.3	3.9	58.0	3.9	54.4	3.9	51.3	3.9	60.5	3.9	0.06

Urban, Transitional, and Nomadic diet patterns account for 20.5%, 10.9%, and 9.8% of variation in intake of pattern components, respectively (total: 41.1%). Factor loadings are generated using annualized average intakes for 334 participants (based on 1834 diet record days); used to calculate season-specific patterns scores for each participant; and seasonal pattern scores are scaled from 0–100. Mean intakes of pattern components within quintiles are also calculated based on annualized averages, and are adjusted for a random intercept per person to account for within-person correlation across seasons. Shading is proportional to magnitude and direction of factor loading (blue: −37.4%, yellow: 0%, red: 73.0%). Bold type indicates *p* < 0.5.

**Table 7 nutrients-12-01514-t007:** Mean diet pattern scores by subgroup-season and province.

Stratum	Urban Pattern Score		Transitional Pattern Score		Nomadic Pattern Score	
	Mean	SE	Mean	SE	Mean	SE
Urban Male Summer	**38.3**	1.6	27.2	0.9	20.6	0.7
Urban Male Winter	**36.1**	1.7	24.0	1.0	21.5	0.7
Urban Female Summer	**27.0**	1.6	**33.5**	0.9	21.5	0.7
Urban Female Winter	23.6	1.7	**32.0**	1.0	20.5	0.7
Rural Male Summer	**25.0**	1.8	24.7	1.0	**27.6**	0.8
Rural Male Winter	**30.9**	1.8	22.2	1.0	**29.1**	0.8
Rural Female Summer	**16.1**	1.8	**28.9**	1.0	**25.3**	0.8
Rural Female Winter	19.6	1.9	**26.0**	1.1	**24.6**	0.8
Ulaanbaatar	37.2	2.1	28.7	1.2	20.7	0.9
Tuv	28.7	2.1	24.6	1.2	23.0	0.9
Bulgan	24.2	2.0	27.7	1.1	21.9	0.9
Khovd	26.2	2.1	28.8	1.2	25.5	0.9
Khuvsgul	26.8	2.0	27.1	1.2	24.9	0.9
Omnogobi	26.5	2.0	29.4	1.1	22.3	0.9
Sukhbaatar	24.5	2.0	26.6	1.2	25.2	0.9
Dornod	26.7	2.1	28.1	1.2	24.7	0.9

Within provinces, means are calculated by first averaging each person’s summer and winter pattern scores and then averaging these within-person annual means across persons in each province. Significant differences in means (*p* < 0.05) based on least squares analysis are indicated for the following planned comparisons: summer vs. winter within urbanicity-sex (indicated in blue), males vs. females within urbanicity-season (green), urban vs. rural within sex-season (bold), and each province vs. the mean of all other provinces (purple) (combinations of colors and/or bolding indicate multiple significant comparisons).

**Table 8 nutrients-12-01514-t008:** Estimated marginal means of diet pattern scores by age.

Urbanicity-Sex	Age Group	Urban Pattern Score		Transitional Pattern Score		Nomadic Pattern Score	
		Mean	SE	Mean	SE	Mean	SE
Urban Male	<33	37.0	1.4	25.6	1.0	19.4	0.7
	33–44	35.2	2.0	24.5	1.4	20.4	1.0
	45+	31.2	2.0	25.3	1.4	22.3	1.0
Urban Female	<33	31.0	1.8	**32.7**	1.3	21.1	0.9
	33–44	29.6	1.7	**32.2**	1.1	23.0	0.8
	45+	25.6	1.7	**34.7**	1.1	21.7	0.8
Rural Male	<33	25.4	2.3	22.0	1.6	24.9	1.1
	33–44	25.3	1.8	23.0	1.2	26.8	0.9
	45+	19.6	1.8	22.8	1.2	28.4	0.9
Rural Female	<33	20.3	2.3	**30.8**	1.6	24.6	1.2
	33–44	22.4	1.7	**27.5**	1.2	26.0	0.8
	45+	20.6	1.9	**26.9**	1.3	26.7	0.9

Means are estimated using a regression model in which summer or winter pattern score (*n* = 635) is predicted using an age group × urbanicity-sex group interaction term adjusted for province, season, within-season mean daily energy intake, and a random intercept per person to account for within-person correlation across seasons. Within urbanicity-sex groups, underlined type indicates a significant linear trend with age (*p* < 0.05). Significant differences in trends are indicated for the following planned comparisons: urban vs. rural within sex (indicated in bold) and males vs. females within urbanicity (green).

**Table 9 nutrients-12-01514-t009:** Estimated marginal means of selected nutrient intakes and clinical measurements by quintile of diet pattern scores.

	Measurement	Energy-Unadjusted										*p*, Trend	Energy-Adjusted										*p*, Trend
		Q1		Q2		Q3		Q4		Q5			Q1		Q2		Q3		Q4		Q5		
		Mean	SE	Mean	SE	Mean	SE	Mean	SE	Mean	SE		Mean	SE	Mean	SE	Mean	SE	Mean	SE	Mean	SE	
**Urban pattern**	Energy (kcal)	1809	117	2181	118	2561	119	2726	125	3391	134	**0.00**											
	Protein (g)	93.5	6.8	110.1	6.8	133.0	6.9	141.5	7.3	178.4	7.8	**0.00**	126.1	4.5	125.0	4.4	130.9	4.4	131.4	4.7	137.9	5.1	**0.01**
	Fiber (g)	5.6	0.6	7.5	0.6	9.6	0.6	11.1	0.6	14.1	0.6	**0.00**	7.7	0.5	8.4	0.5	9.4	0.5	10.5	0.5	11.6	0.5	**0.00**
	Calcium (mg)	859	108	718	109	674	110	636	117	695	125	0.06	1103	106	829	103	660	102	561	109	391	119	**0.00**
	Iron (mg)	11.48	1.04	14.56	1.04	18.10	1.06	19.34	1.12	25.52	1.20	**0.00**	14.85	0.90	16.10	0.87	17.89	0.87	18.28	0.93	21.34	1.02	**0.00**
	Zinc (mg)	13.21	1.02	15.97	1.02	20.04	1.03	21.12	1.10	26.92	1.17	**0.00**	17.17	0.83	17.78	0.81	19.80	0.81	19.91	0.85	21.99	0.94	**0.00**
	Vitamin A (ug retinol eq.))	506	210	753	211	568	213	620	227	1130	243	**0.01**	652	219	817	213	560	212	576	226	951	248	0.61
	Folate (ug DFE)	107	14	134	14	148	14	155	15	205	16	**0.00**	144	13	152	13	145	13	143	14	159	15	0.56
	Vitamin B12 (ug)	9.86	3.26	11.97	3.28	13.85	3.31	13.18	3.53	21.40	3.77	**0.00**	12.26	3.40	13.03	3.30	13.72	3.29	12.44	3.51	18.44	3.85	0.19
	BMI (kg/m^3^)	25.4	0.6	25.5	0.6	25.6	0.6	25.8	0.6	25.9	0.6	**0.00**	25.3	0.6	25.5	0.6	25.6	0.6	25.8	0.6	26.0	0.6	**0.00**
	Ferritin (ug/L)	129.8	7.3	127.2	7.3	123.8	7.3	124.6	7.5	122.3	7.8	0.09	128.3	7.4	126.3	7.4	124.0	7.3	125.1	7.5	124.3	7.9	0.46
	sTfR (mg/L)	4.79	0.38	4.85	0.38	5.06	0.38	5.02	0.39	4.86	0.40	0.55	4.88	0.39	4.90	0.38	5.05	0.38	5.00	0.39	4.75	0.41	0.88
	Iron stores (mg/kg)	10.25	0.38	10.19	0.39	9.85	0.39	9.96	0.39	9.92	0.41	0.08	10.17	0.39	10.14	0.39	9.86	0.38	9.98	0.39	10.03	0.41	0.51
	RBP (umol/L)	1.87	0.09	1.84	0.09	1.90	0.09	1.88	0.09	1.94	0.10	0.28	1.85	0.09	1.83	0.09	1.90	0.09	1.89	0.09	1.97	0.10	0.19
**Transitional pattern**	Energy (kcal)	2865	137	2495	136	2373	136	2329	136	2587	147	**0.01**											
	Protein (g)	134.9	7.9	122.9	7.8	122.2	7.9	122.6	7.8	151.6	8.5	**0.04**	117.5	4.2	123.7	4.2	129.2	4.2	131.5	4.2	147.7	4.5	**0.00**
	Fiber (g)	11.9	0.7	9.1	0.7	8.7	0.7	9.0	0.7	9.2	0.7	**0.00**	10.5	0.5	9.1	0.5	9.2	0.5	9.7	0.5	8.9	0.5	**0.01**
	Calcium (mg)	849	110	718	109	703	109	565	109	763	118	0.14	777	107	721	107	731	108	602	108	748	115	0.41
	Iron (mg)	17.95	1.27	17.59	1.26	16.04	1.26	16.95	1.26	20.11	1.36	0.14	15.76	0.90	17.71	0.90	16.85	0.91	18.09	0.91	19.66	0.97	**0.00**
	Zinc (mg)	19.07	1.20	18.21	1.19	17.72	1.19	18.36	1.19	23.46	1.29	**0.00**	16.70	0.78	18.33	0.78	18.67	0.79	19.56	0.79	22.95	0.84	**0.00**
	Vitamin A (ug retinol eq.))	767	218	801	216	617	217	766	216	620	234	0.44	674	215	806	215	653	218	814	218	602	233	0.73
	Folate (ug DFE)	152	15	150	15	139	15	147	15	158	16	0.79	134	13	151	13	147	13	157	13	153	14	0.07
	Vitamin B12 (ug)	13.73	3.38	13.89	3.35	11.25	3.37	13.86	3.36	17.22	3.64	0.32	12.07	3.33	13.99	3.33	11.89	3.38	14.71	3.38	16.89	3.61	0.12
	BMI (kg/m^3^)	25.9	0.6	25.5	0.6	25.5	0.6	25.6	0.6	25.7	0.6	0.52	25.9	0.6	25.5	0.6	25.5	0.6	25.6	0.6	25.7	0.6	0.54
	Ferritin (ug/L)	125.7	7.4	126.8	7.3	126.0	7.3	127.2	7.4	121.9	7.6	0.41	126.5	7.4	126.9	7.3	125.6	7.4	126.8	7.4	122.2	7.6	0.33
	sTfR (mg/L)	4.98	0.38	5.08	0.38	4.63	0.38	4.90	0.39	5.02	0.39	0.91	4.96	0.38	5.07	0.38	4.64	0.39	4.91	0.39	5.01	0.39	0.98
	Iron stores (mg/kg)	10.08	0.39	10.06	0.39	10.14	0.39	10.06	0.39	9.82	0.40	0.24	10.12	0.39	10.07	0.39	10.12	0.39	10.04	0.39	9.84	0.40	0.18
	RBP (umol/L)	1.86	0.09	1.83	0.09	1.99	0.09	1.86	0.09	1.90	0.10	0.46	1.86	0.09	1.83	0.09	1.99	0.09	1.86	0.09	1.90	0.10	0.45
**Nomadic pattern**	Energy (kcal)	2043	127	2178	121	2359	121	2610	119	3343	128	**0.00**											
	Protein (g)	104.5	7.6	114.7	7.3	123.0	7.3	135.7	7.1	171.8	7.7	**0.00**	126.6	4.8	130.6	4.6	130.6	4.6	131.0	4.4	131.7	4.7	0.25
	Fiber (g)	8.8	0.7	8.9	0.7	8.9	0.7	9.4	0.7	11.5	0.7	**0.00**	10.7	0.5	10.3	0.5	9.5	0.5	9.0	0.5	8.1	0.5	**0.00**
	Calcium (mg)	714	114	688	109	645	109	594	106	937	115	0.15	816	115	761	109	679	108	573	104	756	112	0.12
	Iron (mg)	13.79	1.21	15.23	1.15	16.42	1.16	18.62	1.13	23.84	1.22	**0.00**	16.48	0.99	17.08	0.94	17.27	0.93	18.04	0.90	19.16	0.97	**0.00**
	Zinc (mg)	14.93	1.18	17.11	1.13	18.52	1.13	20.20	1.11	25.38	1.19	**0.00**	17.78	0.91	19.20	0.86	19.47	0.85	19.62	0.83	20.21	0.89	**0.00**
	Vitamin A (ug retinol eq.))	516	225	616	215	589	216	713	211	1092	227	**0.01**	606	234	680	221	619	219	694	211	933	228	0.23
	Folate (ug DFE)	123	15	128	14	139	14	148	14	203	15	**0.00**	147	14	145	13	147	13	144	13	160	14	0.50
	Vitamin B12 (ug)	10.56	3.49	11.58	3.33	11.77	3.34	13.57	3.26	21.77	3.51	**0.00**	12.09	3.61	12.67	3.42	12.27	3.38	13.25	3.26	19.07	3.52	0.09
	BMI (kg/m^3^)	25.8	0.6	25.7	0.6	25.5	0.6	25.6	0.6	25.6	0.6	0.12	25.8	0.6	25.7	0.6	25.5	0.6	25.6	0.6	25.5	0.6	0.07
	Ferritin (ug/L)	123.3	7.5	126.4	7.3	126.3	7.3	124.1	7.3	127.6	7.5	0.54	121.4	7.6	125.2	7.4	125.9	7.3	124.3	7.3	130.8	7.5	0.13
	sTfR (mg/L)	4.77	0.39	5.05	0.38	5.02	0.38	4.97	0.38	4.78	0.39	0.97	4.84	0.40	5.09	0.39	5.03	0.38	4.96	0.38	4.68	0.39	0.58
	Iron stores (mg/kg)	10.04	0.39	10.08	0.39	10.00	0.39	9.93	0.38	10.13	0.40	0.94	9.95	0.40	10.02	0.39	9.98	0.39	9.94	0.38	10.29	0.40	0.34
	RBP (umol/L)	1.87	0.09	1.87	0.09	1.93	0.09	1.87	0.09	1.90	0.10	0.73	1.87	0.10	1.87	0.09	1.93	0.09	1.87	0.09	1.90	0.10	0.69

For each diet pattern, means are estimated using a regression model in which summer or winter mean nutrient intake, biomarker (*n* = 635), or BMI (*n* = 334) are predicted by quintile of pattern score adjusting for age in years, sex, urbanicity, and province; nutrient intake and biomarker models are additionally adjusted for season and a random intercept per person to account for within-person correlation across seasons. Energy-adjusted means are further adjusted for within-season mean daily energy intake. Bold type indicates *p* < 0.5. BMI: body mass index; sTfR: soluble transferrin receptor; RBP: retinol binding protein.

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
