# Peer review of "Diet and Nutrition Status of Mongolian Adults"

_nutrients, 2020, doi:10.3390/nu12051514_

Round 1

Reviewer 1 Report

Thank you for the opportunity to review this paper on diet and nutritional status of adults in Mongolia. This is a unique population and there is limited information available on their diet and nutritional status, despite their increasing risk of chronic disease. Hence, this paper provides important information. Overall, this paper is well-described and I have several comments as below.  

Line 106-107: Serum samples were collected at a random time of day. Please comment potential effects of this approach on biochemical markers the authors have assessed.

Line 120: Please clarify whether CRP is high-sensitivity CRP.

Lines 253-255: The authors have listed nutrients whose inadequacy was prevalent such as calcium and magnesium. However, the discussion of this observation was not included in Discussion section. Please discuss this finding such as by putting it into context of potential reasons for inadequacy of these nutrients; how these nutrient intake levels align with food sources and supply of these nutrients in this population; and how inadequacy of these nutrients may affect prevalence of chronic disease, overweight/obesity and other diseases and health conditions of concern in this population.

Lines 262-263: It is interesting that greater than 10% of rural males in summer had calcium intake above the upper limit, although the inadequacy of calcium is common as described in earlier in this paragraph and Table 2. Without access to Table S5, I was not able to compare tables, but please clarify. Similar to my comment above, please add a discussion on this finding and potential reasons why the calcium intake was particularly high in this group of men and other points, if their calcium intake is indeed high.

Lines 331-332: In the Discussion section, please include potential reasons for this finding of upward trend in BMI by age for urban areas, but not rural areas. In particular, please also mention on physical activity. If this data was not collected in this study, please mention it as a limitation.

Discussion section: Overall, the authors discuss selected topics generally, not specifically, although detailed results on nutrient intake, adequacy and anthropometrics are presented in the Results section. As in the comments above, please include discussions of findings on nutrients other than vitamins A and D. The authors refer to previous studies on the discussion of micronutrient deficiencies in lines 410-414. However, in lines 485-490, the authors mention that different methodologies have resulted in different values and results. Hence, it is of interest to readers that discussion of adequacies and deficiencies of micronutrients other than vitamins A and D reported in this paper is included in this paper.

Author Response

Thank you for the opportunity to review this paper on diet and nutritional status of adults in Mongolia. This is a unique population and there is limited information available on their diet and nutritional status, despite their increasing risk of chronic disease. Hence, this paper provides important information. Overall, this paper is well-described and I have several comments as below.  

Thank you very much for your kind review and thoughtful comments which have helped to improve the paper significantly.

Line 106-107: Serum samples were collected at a random time of day. Please comment potential effects of this approach on biochemical markers the authors have assessed.

Although required for glucose and triglycerides, fasting affects the results of very few other blood tests. Fasting is not required for testing the biomarkers that we measured (ferritin, soluble transferrin receptor, retinol binding protein, and C-reactive protein). In this case, the collection of samples randomly throughout the day was therefore a convenient approach that should not incur error attributable to the timing of meals, and could have the added benefit of avoiding systematic error in biomarkers due to natural diurnal variation (if any) as well as systematic error in dietary measurements that could occur from interrupting all subjects at the same time (e.g. before breakfast). This is helpful to comment on and we have added the following brief clarification to the Methods.

Line 120: Please clarify whether CRP is high-sensitivity CRP.

We have clarified in the Methods that we measured standard CRP.

Lines 253-255: The authors have listed nutrients whose inadequacy was prevalent such as calcium and magnesium. However, the discussion of this observation was not included in Discussion section. Please discuss this finding such as by putting it into context of potential reasons for inadequacy of these nutrients; how these nutrient intake levels align with food sources and supply of these nutrients in this population; and how inadequacy of these nutrients may affect prevalence of chronic disease, overweight/obesity and other diseases and health conditions of concern in this population.

We agree that this is important to elaborate. We have therefore added to the Discussion a summary of the more common nutrient inadequacies and the nutrients of least concern. We have also added likely reasons for these observations, and summarized the potential consequences of observed nutrient inadequacies as well as high observed consumption of unhealthy food groups in terms of national disease burden, citing specific data on the Mongolian food supply and relevant results of epidemiologic studies conducted in and outside of Mongolia.

Lines 262-263: It is interesting that greater than 10% of rural males in summer had calcium intake above the upper limit, although the inadequacy of calcium is common as described in earlier in this paragraph and Table 2. Without access to Table S5, I was not able to compare tables, but please clarify. Similar to my comment above, please add a discussion on this finding and potential reasons why the calcium intake was particularly high in this group of men and other points, if their calcium intake is indeed high.

We are sorry that it was not possible to access the Supplementary Tables. High calcium intake was more prevalent among urban males in the summer mainly because their mean total dairy consumption was very high (657g/day, which was the highest mean dairy intake observed among the four population subgroups in both seasons) (Table 1). Sixty-two percent of calcium intake among rural males in the summer was contributed by milk (including fermented mare’s milk), milk tea, milk soups (Table S7). These findings are not very surprising to us because milk production among nomadic families in Mongolia known to be highly seasonal with a peak in the summertime [1], however, they are significant and we agree that they would be helpful to mention, and we have done so in the Results and the Discussion.

[1] FAO/UNICEF/UNDP. Joint Food Security Assessment Mission to Mongolia; Food and Agriculture Organization of the United Nations: Ulaanbaatar, Mongolia, 2007.

Lines 331-332: In the Discussion section, please include potential reasons for this finding of upward trend in BMI by age for urban areas, but not rural areas. In particular, please also mention on physical activity. If this data was not collected in this study, please mention it as a limitation.

We agree this is a key point that should be discussed further. As shown in Figure 4 Panel B, the urban-rural difference in age trends in BMI cannot be explained by differences in energy intake. One likely reason for this difference may be the intense physically-active lifestyle that many nomadic Mongolians continue to engage in even into older age, while lifestyles in urban areas have become increasingly sedentary [1]. Physical activity was not measured in this study and we have listed this as an important limitation as per the Reviewer’s suggestion. A second likely reason for this difference may be related to our finding that the Urban dietary pattern on the whole appears to be more obesogenic than the Nomadic pattern (Figure 4 Panel D), a finding that robust to age and energy adjustment. Analysis of large prospective studies in the U.S. have shown that increased intake of specific foods often associated with industrialized and westernized diet patterns globally (such as refined grain products and sugar-sweetened beverages) predispose people to weight gain irrespective of increases in energy intake [2]. We have added these points in a new Discussion paragraph.

[1] World Health Organization. Mongolian STEPS Survey on the Prevalence of Noncommunicable Disease and Injury Risk Factors-2013; World Health Organization: Ulaanbaatar, Mongolia, 2013.

[2] Mozaffarian, D.; Hao, T.; Rimm, E.B.; Willett, W.C.; Hu, F.B. Changes in diet and lifestyle and long-term weight gain in women and men. N Engl J Med. 2011, 364, 2392–2404.

Discussion section: Overall, the authors discuss selected topics generally, not specifically, although detailed results on nutrient intake, adequacy and anthropometrics are presented in the Results section. As in the comments above, please include discussions of findings on nutrients other than vitamins A and D. The authors refer to previous studies on the discussion of micronutrient deficiencies in lines 410-414. However, in lines 485-490, the authors mention that different methodologies have resulted in different values and results. Hence, it is of interest to readers that discussion of adequacies and deficiencies of micronutrients other than vitamins A and D reported in this paper is included in this paper.

The Reviewer’s excellent comments are well-noted. To help readers understand the significance and context of key findings, and make the Discussion more specific, we have expanded the discussion of adequacies and inadequacies by two paragraphs, in line with the Reviewer’s earlier comment. (As noted in the limitations section, differences between the current study and our previous work affect two nutrients in certain population subgroups, and do not create qualitatively different conclusions.)

Reviewer 2 Report

This is a truly excellent manuscript describing a fascinating and globally relevant study. I now will turn to the authors' earlier papers to read more! My only recommended change is to ask that as authors , and I will do the same as a reviewer, if at the end of your manuscript on line 509 where the location of the Supplementary Materials is provided, that you actually list the titles of the 7 Supplementary tables for the readers. I believe that this will further encourage the readers to turn to these tables which provide in depth information about the work you have done.  

Again - congratulations of superb work and this manuscript!

Author Response

This is a truly excellent manuscript describing a fascinating and globally relevant study. I now will turn to the authors' earlier papers to read more! My only recommended change is to ask that as authors , and I will do the same as a reviewer, if at the end of your manuscript on line 509 where the location of the Supplementary Materials is provided, that you actually list the titles of the 7 Supplementary tables for the readers. I believe that this will further encourage the readers to turn to these tables which provide in depth information about the work you have done.  

Again - congratulations of superb work and this manuscript!

We sincerely thank the Reviewer for their thoughtful review and very kind comments on our study. As suggested, we have listed the titles of the seven supplementary tables in the Supplementary Materials section, and we agree this is a helpful idea to better guide the readers.

Reviewer 3 Report

The research has been conducted very well, both in terms of methodology as well as statistics, on a considerably large group of Mongolian adults. Admittedly, the sample was not nationally-representative but both the size and breadth of the study population (of summer and winter prospective dietary intake from urban and rural men and women in 8 national provinces) and the use of random sampling within each province, this, however, this study has been the most study of the diet among Mongolian adults.

What I would like to particularly emphasize is the manuscript, written in a clear manner, easy to read and understand for the reader.

Strengths and limitations – well discussed.

Appropriate bibliography has been used in the text of the article.

However, I would like to point out some minor issues regarding the content:

Abstract: line 30-34, I suggest to clarify - the data was collected twice from the same people.

in point 2.1. The data was collected for 5/6 years (from 2011 to 2016) did it affect the results obtained?

in point 4. Discussion (line 397-398): "The present study found a high prevalence of key dietary nutrient inadequacies in a nationwide sample of Mongolian adults in summer and winter, particularly those of fiber, folate, and vitamin D." - what about vitamin C and E deficiency? - table 2.

Additionally, I suggest separating parts: Strengths and limitations and Conclusion.

The paper is an interesting addition to the literature and worthy of special publicity.

Thank you for allowing me to review your study.

Author Response

The research has been conducted very well, both in terms of methodology as well as statistics, on a considerably large group of Mongolian adults. Admittedly, the sample was not nationally-representative but both the size and breadth of the study population (of summer and winter prospective dietary intake from urban and rural men and women in 8 national provinces) and the use of random sampling within each province, this, however, this study has been the most study of the diet among Mongolian adults.

What I would like to particularly emphasize is the manuscript, written in a clear manner, easy to read and understand for the reader.

Strengths and limitations – well discussed.

Appropriate bibliography has been used in the text of the article.

We thank the Reviewer very much for their kind review of our study, and helpful comments.

However, I would like to point out some minor issues regarding the content:

Abstract: line 30-34, I suggest to clarify - the data was collected twice from the same people.

We agree, and have clarified in the Abstract that 3-day diet records were collected twice from each participant in summer and winter, and anthropometry was collected once from each participant.

in point 2.1. The data was collected for 5/6 years (from 2011 to 2016) did it affect the results obtained?

We thank the Reviewer for this very important point. Rapid trends in urbanization and global market integration that have occurred in Mongolia in recent years have probably had effects on the diet. This would be an interesting question to examine. However, due to resource limitations, each year of the study involved data collection from up to three provinces only, and in each province, collection was completed over the course of one year. Therefore, any analysis of trends from 2011 to 2016 would be intractably confounded by region. This is a limitation of the study, and we have included mention of it in the Strengths and Limitations section. In our conclusion, we also stress the importance of developing national dietary surveillance in Mongolia, as this will be necessary to understand time trends.

in point 4. Discussion (line 397-398): "The present study found a high prevalence of key dietary nutrient inadequacies in a nationwide sample of Mongolian adults in summer and winter, particularly those of fiber, folate, and vitamin D." - what about vitamin C and E deficiency? - table 2.

We agree that these inadequacies are important to mention. We have altered the structure of the Discussion so that the first paragraph covers three nutrients that have been studied more extensively in prior Mongolian surveys (vitamin D, vitamin A, and iron), followed by a new paragraph that includes a list of other major observed inadequacies (fiber, calcium, magnesium, thiamin, folate, and vitamins B6, C, D, and E), likely reasons for these other inadequacies, and potential consequences for national disease burden.

Additionally, I suggest separating parts: Strengths and limitations and Conclusion.

We agree, and have separated the Conclusion section from the rest of the Discussion.

The paper is an interesting addition to the literature and worthy of special publicity.

Thank you for allowing me to review your study.

Thank you again for your very kind and helpful review.